# Inhibitory microcircuits for top-down plasticity of sensory representations

Katharina Anna Wilmes[1] & Claudia Clopath [1]*

Rewards influence plasticity of early sensory representations, but the underlying changes in circuitry are unclear. Recent experimental findings suggest that inhibitory circuits regulate learning. In addition, inhibitory neurons are highly modulated by diverse long-range inputs, including reward signals. We, therefore, hypothesise that inhibitory plasticity plays a major role in adjusting stimulus representations. We investigate how top-down modulation by rewards interacts with local plasticity to induce long-lasting changes in circuitry. Using a computational model of layer 2/3 primary visual cortex, we demonstrate how interneuron circuits can store information about rewarded stimuli to instruct long-term changes in excitatory connectivity in the absence of further reward. In our model, stimulus-tuned somatostatin-positive interneurons develop strong connections to parvalbumin-positive interneurons during reward such that they selectively disinhibit the pyramidal layer henceforth. This triggers excitatory plasticity, leading to increased stimulus representation. We make specific testable predictions and show that this two-stage model allows for translation invariance of the learned representation.

[1] Bioengineering Department, Imperial College London, SW72AZ London, UK. *email: c.clopath@imperial.ac.uk

Animals learn better when it matters to them. For example, they learn to discriminate sensory stimuli when they receive a reward. As a result of learning, neural responses to sensory stimuli are adjusted even in primary sensory areas, such as primary visual cortex (V1, refs. [1–3]). When mice consistently receive a reward after seeing a grating of a given orientation, the tuning preference of layer 2/3 neurons for this rewarded orientation is increased[1,3]. It is thought that behaviourally relevant stimuli, such as rewards, trigger an internal top-down signal available to these early sensory circuits. This could be mediated by cholinergic inputs from the basal forebrain, for example (see refs. [4,5]). By top-down signal, we mean any long-range input to the superficial layers that delivers behaviourally relevant information to the local circuit.

Pyramidal cells in primary sensory cortices are embedded in a canonical microcircuit motif with different types of inhibitory interneurons. The main inhibitory types are the PVs, the SSTs and the VIPs. Top-down inputs project to superficial layers[6–11]. They target multiple cell types. For example, VIPs in the primary auditory cortex are activated when a reward is present[9]. Inhibitory synapses are plastic (see ref. [12] for a review) and perturbation of interneurons impairs learning ([4,13], see ref. [14] for a recent review).

We hypothesised that the inhibitory circuitry in layer 2/3 mediates the top-down instructions (e.g. triggered by a reward) to guide plastic changes in the circuit beyond the presence of reward. We wanted to test whether interneurons can learn from a top-down signal in a model of cortical circuitry. The inhibitory connectivity structure could then instruct the excitatory cells in the absence of top-down modulation. To test this, we built a biologically constrained computational model of layer 2/3 primary visual cortex. We simulated a rewarded phase in which the presentation of one stimulus is paired with a reward signal, which excites VIPs. We then simulated a refinement phase, where the sensory stimuli were presented without the reward. During the first rewarded phase, connections between SSTs and PVs developed a specific connectivity structure. This structure triggered disinhibition of the excitatory neurons even in the absence of reward. Plasticity in the excitatory neurons, therefore, shaped the microcircuit during the second refinement phase. It led to an increased stimulus preference of the previously rewarded stimulus. Our model offers testable predictions on the activity of different cell types during and after the reward presentation. We also propose that this two-stage mechanism allows for learned representations to generalise across different parts of the visual space.

## Results

### Two-stage model of top-down guided microcircuit plasticity.
Neural responses to visual stimuli in V1 are not a simple function of bottom-up sensory inputs. They are additionally modulated by various inputs from other areas[15,16] and by recurrent local excitatory and inhibitory neurons (especially in layer 2/3, ref. [17]). We hypothesised that top-down inputs can induce changes in sensory representations via changes in recurrent connections in two stages.

(i) The rewarded phase. A specific stimulus (e.g. a vertical bar) is paired with a reward-mediated top-down signal which excites VIPs (triple arrow in Fig. 1a top). The VIPs inhibit the SSTs, which we assume are stimulus-tuned[18,19]. At the same time, the VIPs disinhibit the PVs, which we model as untuned[11,18–20]. Activity-dependent plasticity then increases the connection strengths between SSTs which are tuned to the rewarded stimulus (here the vertical bar, vertSSTs) and PVs (Fig. 1b top). The inhibitory motif now carries information about the reward. In

addition, this inhibitory structure disinhibits the excitatory neurons (Fig. 1b bottom).

(ii) The refinement phase. In the second phase, the reward and therefore also the top-down input is absent (Fig. 1b top). As the inhibitory (SST-PV) motif disinhibits the PCs (Fig. 1b top), it opens a window for plasticity at the excitatory synapses. This will result in a refinement of the excitatory connectivity. Strong recurrent connections from PCs coding for the vertical bar to other excitatory neurons will develop. All PCs will, therefore, have an increased response to the vertical bar stimulus.

In summary, we hypothesised that learning can happen in two stages. To test this, we simulated a mechanistic model of the layer 2/3 microcircuit.

### Reward signal triggers plasticity in the inhibitory circuit.
We simulated a spiking neural network model of the canonical microcircuit of layer 2/3 mouse primary visual cortex[21]. Neurons were modelled as integrate-and-fire neurons. Connection probabilities and strengths were constrained by experimental data[20–24]. VIPs inhibited the SSTs, which in turn inhibited all other cell types. The PVs inhibited the PCs and themselves. The PCs were recurrently connected and excited all interneuron types (Fig. 1)[21]. PCs and SSTs were tuned to orientation[18,19], but see ref. [25]. PVs were coupled via gap junctions[26]. Recurrent excitatory connections and those from SSTs to PVs were plastic according to the classical spike-timing-dependent plasticity (STDP) model. Unless we explicitly state that all connections were plastic (as in Supplementary Fig. 4), the other connections were fixed (see Methods section for details).

Before we tested our hypothesis, we needed to bring our model from random initial connectivity to a set of weights that corresponds to adult V1 connectivity. We call that the developmental phase. During this phase, we randomly presented inputs to our network corresponding to oriented gratings. Excitatory neurons that code for the same orientation were coactive. Therefore, they formed strong clusters due to Hebbian learning[27,28] (Fig. 2f developmental phase, Fig. 2e middle). The SST-to-PV weights did not form a specific structure during the developmental phase (Fig. 2b developmental phase) consistent with experimental literature[29].

We then simulated a rewarded phase (grey background in Fig. 2b and f). A reward signal excited the VIP population when the vertical bar stimulus was present. This top-down signal was in itself untuned. However, the temporal coincidence with the vertical bar made it stimulus-specific. Connections from the vertical-bar tuned SSTs to PVs increased (purple line in Fig. 2b, rewarded phase). The resulting SST-to-PV structure (Fig. 2d) carried information about the identity of the rewarded stimulus. Hence, the PVs became less responsive to the rewarded stimulus (vertical bar, Fig. 2c). Notably, no significant stimulus-specific structure arose between excitatory connections (Fig. 2e and f). Accordingly, the tuning of excitatory populations did not change. Note that an excitatory structure could also co-develop with the inhibitory structure (Supplementary Fig. 3), depending on the disinhibition of PCs during the reward.

In summary, unspecific top-down signals can induce an inhibitory connectivity structure without changing excitatory connectivity.

### Inhibitory structure guides plasticity in absence of reward.
We then tested whether the interneuron structure can guide plasticity in the excitatory neurons in the absence of reward. After the rewarded phase, the inhibitory structure effectively disinhibited all PCs when a vertical bar was present (corresponding to the previously rewarded stimulus). The vertically tuned PCs fired a

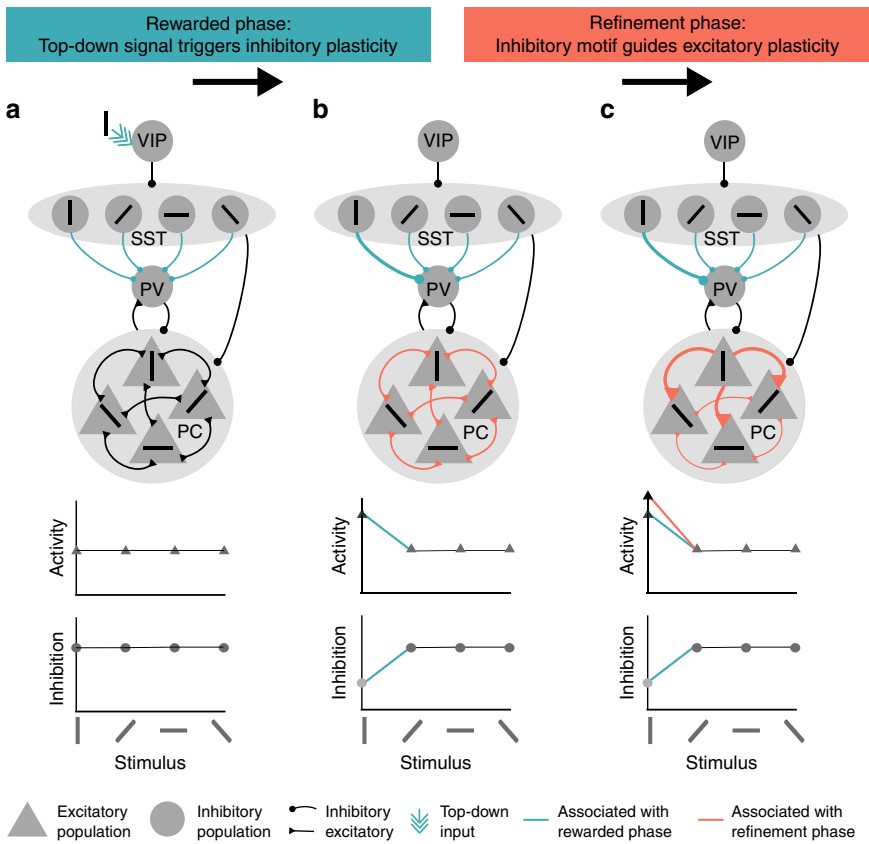

**Fig. 1** The two-stage model of top-down guided plasticity. **a** Before the rewarded phase. We assume the SSTs and PCs to be stimulus-tuned and PVs to be untuned. During the rewarded phase, the top-down signal activates VIPs when the rewarded stimulus is present (vertical bar). This triggers plasticity at the SST-to-PV connections. **b** At the end of the rewarded phase and at the beginning of the refinement phase, there are strong connections from the SSTs tuned to the vertical bar to PVs (green, top). The PV activity is therefore low for the vertical bar (bottom). The excitatory neurons coding for the vertical bar are disinhibited (middle). During the refinement phase, the inhibitory motif guides plasticity at the excitatory neurons. **c** At the end of the refinement phase, strong connections from the excitatory neurons coding for the vertical bar to the other excitatory neurons have developed (red, top). This results in an increased activity of excitatory neurons towards the vertical bar (red line, middle) beyond that resulting from reduced inhibitory PV activity (blue line, middle, and bottom). Note that existing connections between cell types are omitted in the figure to increase clarity

few milliseconds before the other PCs because they received additional feedforward inputs. STDP, therefore, led to a strengthening of the connections from the vertically tuned PCs to the other PCs (Fig. 3b purple line). Accordingly, connections in the reverse direction were depressed (Fig. 3b green line; see Supplementary Fig. 1 for spiking details). As a result of the excitatory connectivity structure (Fig. 3c), all PC populations showed an increased response to the vertical bar (Fig. 3d). Since PCs are driving the PVs, PVs became tuned to the vertical bar (Fig. 3h). Synergistically, SST-to-PV connections were strengthened even further (Fig. 3f).

Note, the excitatory structure was stable even if we artificially deleted the inhibitory structure (Supplementary Fig. 8). Finally, we also showed that precise spike timing was not necessary for our two-stage model (see a rate-based implementation in Supplementary Fig. 10).

In summary, the inhibitory network structure can induce changes in sensory representation by guiding excitatory plasticity.

**Experimentally testable predictions**. Our model makes eight precise experimentally testable predictions. (1) PCs become more tuned to the rewarded stimulus (Fig. 3d). (2) PVs initially become less tuned to the rewarded stimulus (Fig. 2c), but eventually become more tuned to the rewarded stimulus when the excitatory structure develops (Fig. 3h). (3) SSTs also slightly increase their response to the rewarded stimulus (Supplementary Fig. 2) and (4)

VIPs do not change their tuning. (5) Both PC and (6) PV firing rate responses to the rewarded stimulus increase relative to other stimuli (Fig. 4a, b). (7) Excitatory and inhibitory currents increase during the rewarded stimulus (Fig. 4d, Supplementary Fig. 10g). (8) The E/I ratio increases in some cells during the rewarded stimulus (Fig. 4e).

Furthermore, although technically hard to test experimentally, we predict that blocking excitatory plasticity during the task will not abolish learning, whereas blocking inhibitory plasticity will. However, blocking excitatory plasticity after the task, i.e. during the refinement phase, will abolish learning.

**Translation invariance of learned representations**. Most excitatory neurons in layer 2/3 are simple cells[30], which respond to dedicated visual field locations. Changes in connectivity between these cells will hence only affect the representation of a stimulus at that visual field location. Therefore, we wondered whether and how the increased representation of the rewarded stimulus could generalise to visual field locations that were not rewarded. In particular, we asked whether learning the inhibitory structure can lead to enhanced stimulus representations that are invariant to the visual field location. This so-called translation invariance is a general property of the visual system. For example, how we perceive an edge should be independent of where in the visual field it occurs.

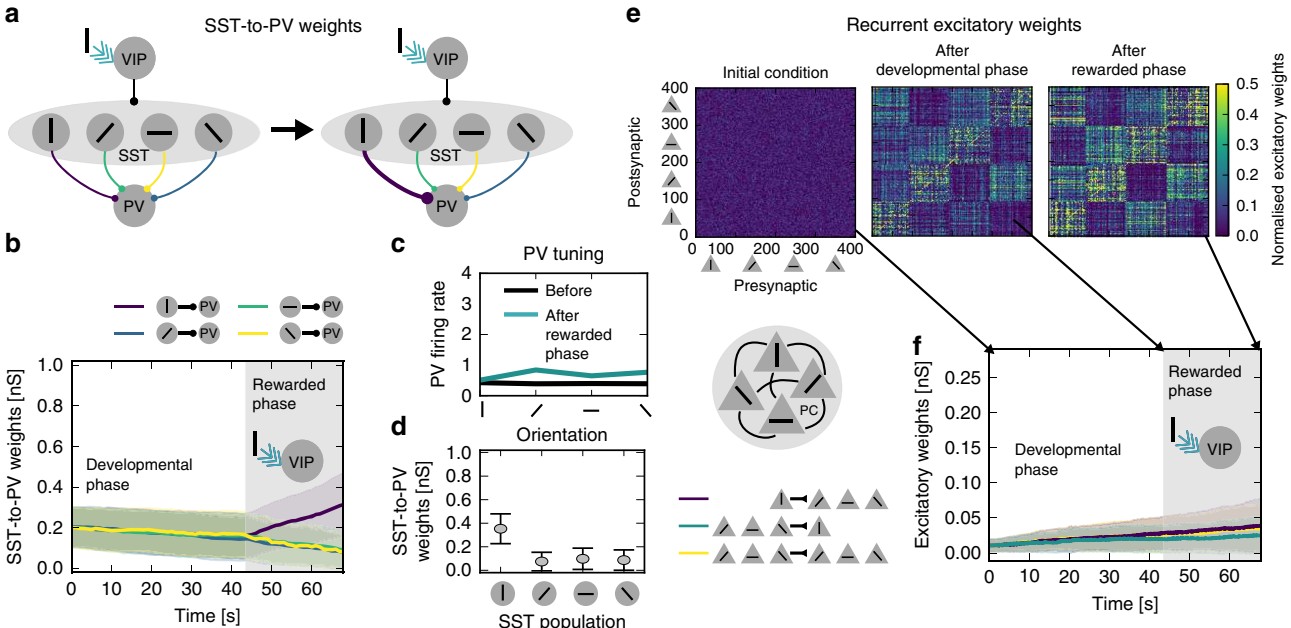

**Fig. 2** Top-down reward signal triggers plasticity in the inhibitory circuit. **a** Illustration of changes in the inhibitory structure during top-down modulation (cyan arrow). **b** Evolution of the inhibitory SST-to-PV connections, grouped according to SST tuning preference (colours match the connections in **a**, shown are the mean and s.d.). **c** Tuning of PVs after the rewarded phase. **d** SST-to-PV weights (mean and s.d.) at the end of the rewarded phase, grouped according to SST tuning preference. **e** Recurrent excitatory weights at different time points in the simulation (initial random condition, after the developmental phase, after the rewarded phase). **f** Evolution of excitatory connections (mean and s.d.). Vertically tuned PCs to vertically tuned PCs (green), vertically tuned PCs to non-vertically tuned PCs (purple), non-vertically tuned PCs to non-vertically tuned PCs (yellow)

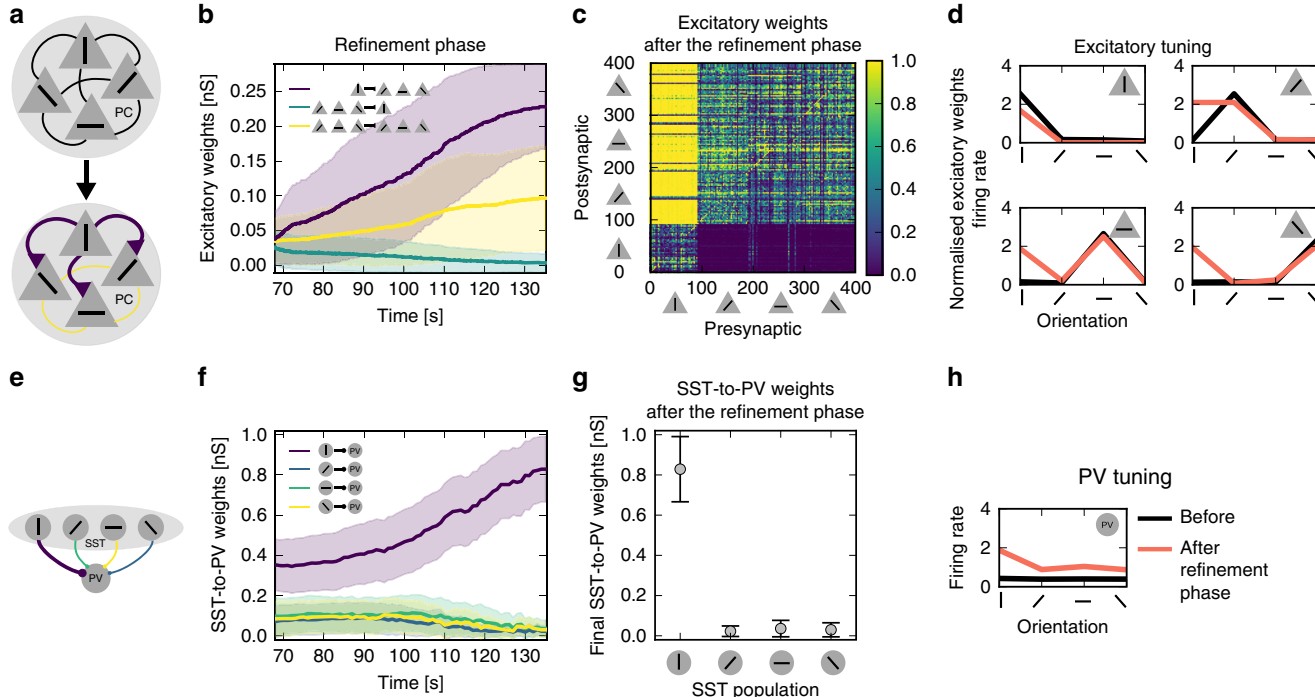

**Fig. 3** Inhibitory structure guides excitatory plasticity in the absence of reward. **a** Illustration of changes in the excitatory structure. **b** Evolution of excitatory connections. Mean and s.d. of the connections from the vertically tuned PCs to PCs tuned to other orientations (purple), from PCs tuned to other orientations to the vertically tuned PC population (green), and from others to others (yellow). **c** Final excitatory weight matrix. Neurons 1–100 are tuned to the vertical bar, 100–200 to an angled bar, etc. **d** Tuning of excitatory populations before the rewarded phase (black) and at the end of the refinement phase (red) (number of spikes during 50 ms after stimulus onset averaged over all occurrences of that stimulus in 1 s of simulation). **e** Illustration of the inhibitory structure after the rewarded phase. **f** Evolution of the SST-to-PVs connections (mean and s.d.), grouped according to SST tuning (colours match the colours of the connections in **e**). **g** SST-to-PV connections after the refinement phase, grouped by SST tuning (error bars: s.d.). **h** Tuning of PVs after the refinement phase

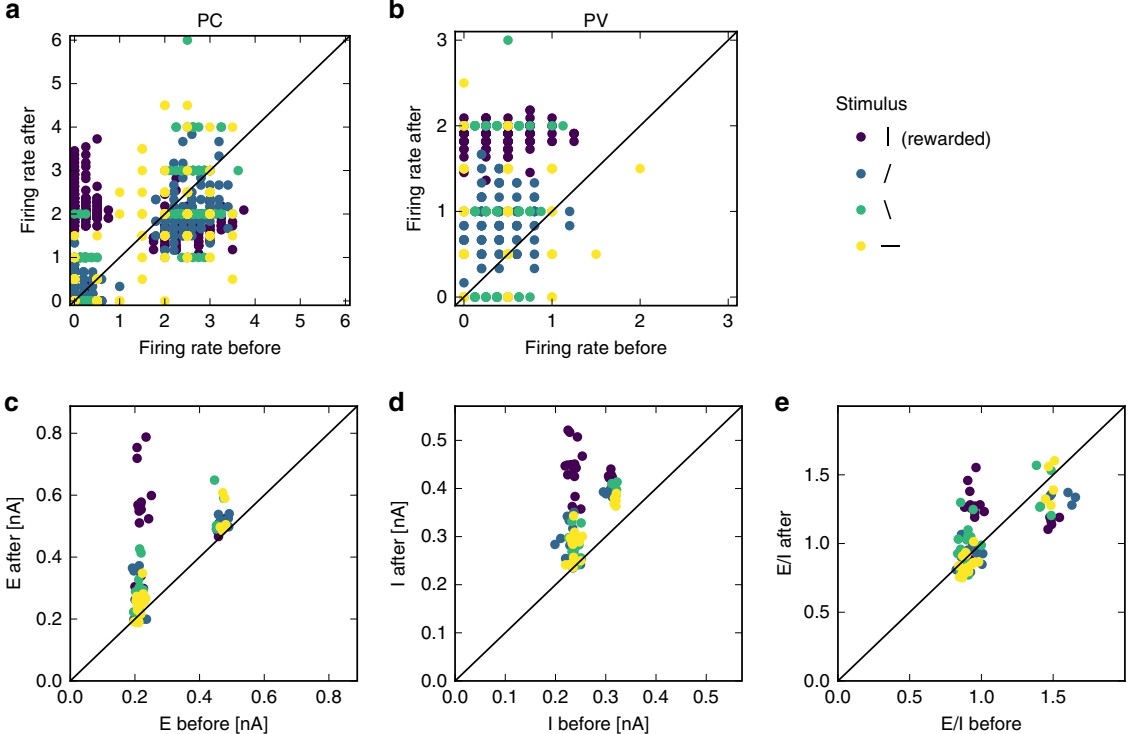

**Fig. 4** Predictions of the model. **a** Firing rate of 20 sample excitatory neurons (5 from each population) before the rewarded phase as a function of after the refinement phase. **b** Firing rate of sample PV interneurons. **c** Excitatory currents (E), **d** Inhibitory currents (I) and **e** E/I ratio of the same 20 sample excitatory neurons

To test this, we expanded our model to include another set of PCs, tuned to the same orientations but to a different visual location. All PCs in the model were innervated by the same set of interneurons, which were tuned to both locations (Fig. 5a) (assuming inhibition with broader spatial receptive fields, see ref. [31]). As before, SST-to-PV structure developed during the rewarded phase (Fig. 5e). In the refinement phase, the excitatory structure emerges in both PC subnetworks (Fig. 5b, h, and c, i), leading to an increased representation of the stimulus for both visual field locations (Fig. 5d and j). Note that the extent of the generalisation is directly dependent on the spatial tuning width of the interneurons, and on the connectivity from the interneurons to the non-rewarded set of PCs. In summary, our two-stage model allows for a generalisation of the learned representation to other visual field locations, under the assumption that interneurons are spatially broadly tuned and project broadly.

## Discussion

We propose that a memory of the rewarded stimulus is stored in the inhibitory structure. It can instruct excitatory plasticity in the absence of reward via a disinhibition mechanism. The PCs then increase their tuning to the rewarded stimulus because they receive strong connections from PCs coding for the rewarded stimulus, regardless of their initial tuning (see also Supplementary Fig. 6).

We show that an unspecific top-down reward signal is sufficient to create a specific circuit structure owing to the temporal coincidence between reward signals and stimulus-evoked activity. Where does the top-down signal come from? One candidate is cholinergic fibres from the forebrain, which have been shown to modulate activity in V1[5,32]. In addition, the nucleus basalis in the basal forebrain, which sends widespread cholinergic projections to all sensory areas, has long been known to play a role in cortical map plasticity[33], learning, and memory. Lesioning and applying cholinergic antagonists impair learning and memory[34]. Nucleus basalis stimulation and local ACh administration alter auditory receptive fields[35,36]. Finally, cholinergic inputs are involved in experience-dependent plasticity of visual cortex[37]. Cholinergic inputs target many interneuron cell types[38], and continue to drive VIP activity into adulthood[39]. Here we focused on top-down modulation of VIPs, as VIPs (i) directly respond to reinforcement signals[9], (ii) inhibit other interneurons during learning[4,6], and (iii) are diversely modulated by glutamatergic, cholinergic and serotonergic inputs[10,40]. Finally, another likely candidate are higher-order thalamic inputs as they increase VIP-mediated disinhibition of PCs and PVs, and thereby gate synaptic plasticity[41].

Our model requires one tuned interneuron type to store the identity of the rewarded stimulus. The stimulus selectivity of interneurons is debated. Most data on interneuron tuning comes from studies identifying interneurons with GAD65-GFP or GAD67-GFP, which are non-specific GABAergic interneuron markers that do not distinguish between different cell types (such as SSTs and PVs). Those studies report a broad tuning of interneurons in layer 2/3 of the mouse[25,30,31,42]. Studies looking at PV interneurons in particular tend to find similarly broadly tuned response properties[11,20], but see ref. [43], where some PV cells were sharply tuned. Runyan et al.[44] report that PVs can both be sharply or broadly tuned but are generally less tuned than PCs. There are few studies investigating the tuning of SSTs[18,19]. These indicate that SSTs are more sharply tuned than PVs. SSTs are odour-tuned in piriform cortex, and also seem to be more frequency-tuned than PVs in auditory cortex[45]. Therefore, SSTs could provide the tuning properties needed for the proposed mechanism. Importantly, the proposed mechanism does not depend on the tuning properties of PV cells as our simulations yield similar results for both untuned and tuned PV cells (Supplementary Fig. 5).

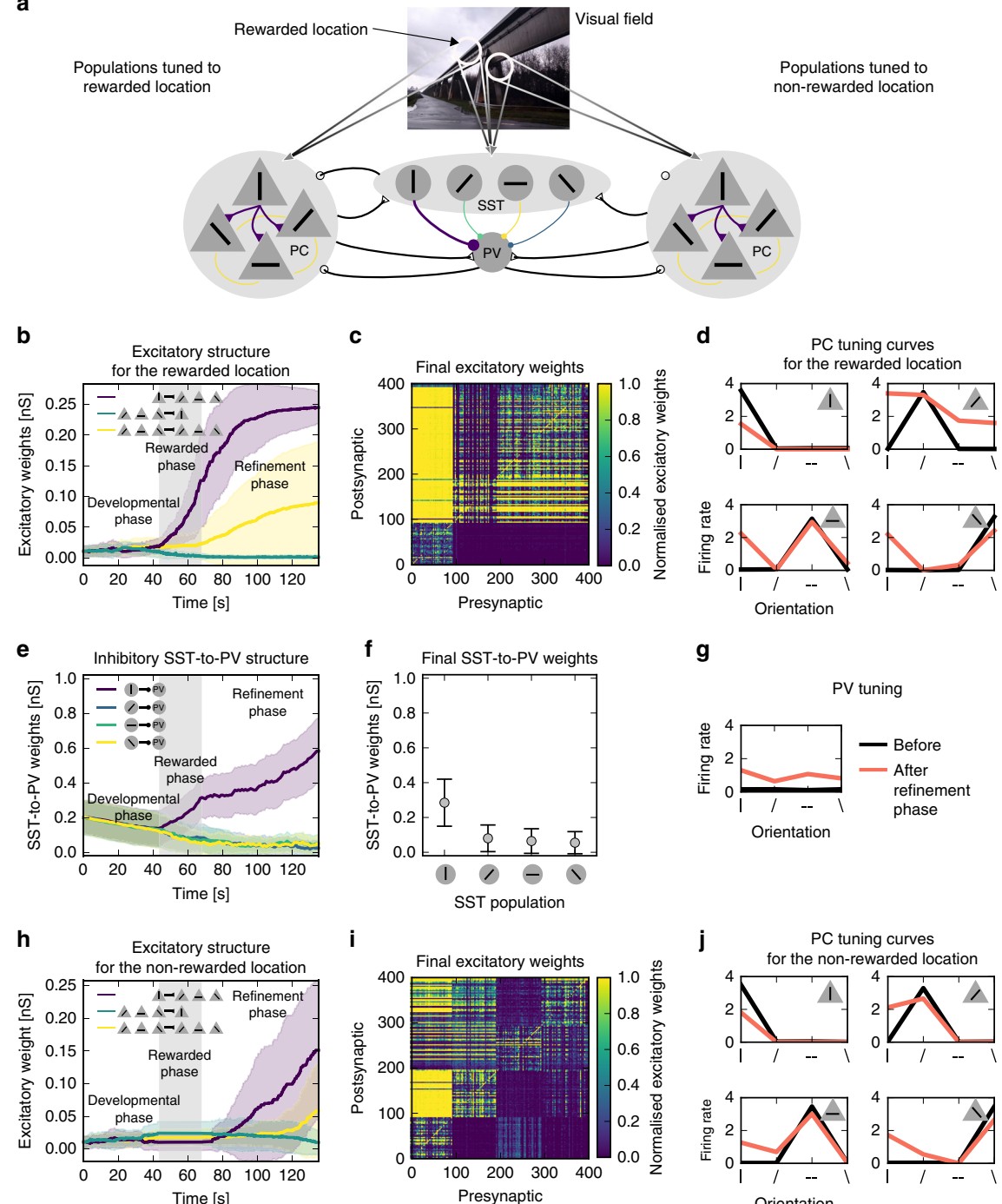

**Fig. 5** Translation invariance of learned representations. **a** Illustration: Two excitatory networks with different visual receptive field locations (white circles) share the same interneuron network. The interneurons are broadly tuned, and receive input from both visual field locations. Only one visual field location is rewarded during the rewarded phase (rewarded location). **b** Evolution of excitatory weights for the rewarded location. **c** Final excitatory weights for the rewarded location 1. **d** Tuning curves of excitatory populations with a receptive field at the rewarded location at the beginning (before) and at the end of the simulation (after the refinement phase; measured as the number of spikes during 50 ms after stimulus onset averaged over all occurrences of that stimulus in 1 s of simulation). **e** Evolution of inhibitory synaptic weights. **f** Final inhibitory weights. **g** PV tuning at the beginning and after the refinement phase. **h** Evolution of excitatory weights for the non-rewarded location. **i** Final excitatory weights for the non-rewarded population. **j** Tuning curves of excitatory populations with a receptive field at the non-rewarded location (as in **d**)

In Goltstein et al.[1] and Goltstein et al.[46], the broadening of tuning curves was restricted to cells which were tuned to a similar orientation as the rewarded stimulus. Our model can capture this finding under the condition that the connectivity between pyramidal cells is restricted based on the distance between their preferred orientations (Supplementary Fig. 6).

In Khan et al.[2], the majority of cells increased their selectivity for one stimulus by selectively suppressing their response to the other stimulus. This was not the case in Poort et al.[3], where cells both increased and decreased their responses. In Goltstein et al.[1] and Goltstein et al.[46], cells simply increased their response to the rewarded stimulus by broadening their tuning curves. The

possible discrepancy may arise from the design of the studies. Whereas Poort et al.[3] and Khan et al.[2] quantified responses to the two task-relevant stimuli, Goltstein et al.[1] calculated tuning curves for a range of orientations including two task-relevant orientations. In our model, we captured the increased stimulus representation by an increase of responses to the rewarded stimulus, which was observed in all experimental studies[1–3]. The underlying changes likely involve a mixture of strengthening and weakening of connections (Supplementary Fig. 7). In addition, PVs were also shown to increase their selectivity with learning[2].

Our point neuron model does not capture the fact that SSTs and PVs target different dendritic regions. However, we do not expect our results to change if we include dendrites in our model, as the PV-mediated disinhibition projects somatically.

It would be interesting to study the effect of multiple modulatory inputs, such as long-range glutamatergic, serotonergic, and cholinergic inputs. How do these inputs interact? Do they interfere with each other? How are different signals distinguished? For instance, both learning and attention affect the selectivity of responses of neurons in the circuit[2,3].

We assume that the change in representation results from local changes in the circuit. Alternatively, it could result from changes in top-down or bottom-up influences. Learning an association between an aversive stimulus and a grating, for example, was reported to be correlated with increased top-down inputs from the retrosplenial cortex[47]. Top-down influences alone, however, do not account for the persistence of the changes during anaesthesia[1,3]. Learning could involve a multitude of adjustments in recurrent, bottom-up and top-down connectivity, including higher thalamocortical inputs[48].

What happens to the increased representation when the environment changes? In our model, inhibitory and excitatory structures reinforce each other, which is beneficial for maintaining the memory. In other situations, however, it may be useful to unlearn the structure. For example, if another stimulus is paired with reward. It may also be possible to maintain multiple representations simultaneously, depending on the capacity of the network. Synaptic competition could naturally select a winner if capacity is reached. Another example is extinction. Animals can unlearn an association, which can be correlated with losing the increased representation of the rewarded stimulus[49]. The mechanism underlying extinction is, however, unknown. In the amygdala for example, the neural correlate of extinction is not a simple reversal of the changes that happened during learning, but rather a re-learning mechanism[50]. This is in line with studies showing that extinction is context-dependent and extinct associations can be unmasked (for a review see ref. [51]). It is therefore possible that the increased low-level representation is also maintained beyond extinction.

In our model, the development of the excitatory structure during the refinement phase requires neural activity. We therefore continued to randomly present different stimuli without pairing them with a reward signal. The question is whether this unpairing should induce extinction of the learned association rather than consolidation. Interestingly, in the study by Grewe et al.[50] during the presentation of the CS alone after pairing, there was no immediate extinction, but actually a strengthening of the changes that happened during pairing. This supports a different underlying mechanism for extinction.

We list below alternative mechanisms that would result in an increased stimulus representation. We argue, however, that our model is the one most in line with experimental data from the visual cortex.

(i) Vertically tuned PCs may develop strong connections to VIPs. It will inhibit SSTs and therefore disinhibit PCs. This motif, however, will cause VIPs to become more tuned during learning, which was not observed experimentally[2].

(ii) Vertically tuned PCs may develop strong connections to SSTs. It will inhibit PVs and therefore disinhibit PCs. This motif will result in a tuning increase of SSTs, which was also not observed experimentally[2].

(iii) Vertically tuned PCs may reduce their inhibition of PVs, thereby increasing the activity of all PCs. This can lead to instability and contradicts the finding that PCs and PVs increase their effective connectivity during learning[2].

(iv) SSTs may decrease their response to the rewarded stimulus more than to other stimuli. This motif predicts a change in the tuning of SSTs, which does not seem to be the case in Khan et al.[2].

We propose three benefits of an intermediate inhibitory structure over direct changes in recurrent excitatory connectivity. (i) Bridging timescales: The reward is only present for a short amount of time, but plasticity can be slow. These two timescales can be bridged because the inhibitory and excitatory structure mutually reinforce each other. Therefore, a strong excitatory structure can emerge beyond the presence or even in the absence of reward. In addition, high inhibitory firing rates, typical of PVs, could effectively increase the inhibitory learning rate, allowing for a rapid development of the inhibitory structure. (ii) Translation invariance: We showed that the inhibitory structure allows for the increased representation to generalise across visual locations. Interestingly in machine learning, translation invariance which improves generalisation is achieved by weight sharing. The same weight vector (filter) is applied to different regions of the input space. This has been considered biologically implausible, as synaptic weights of other synapses are not locally available to each synapse. Broadly tuned interneuron networks that are shared across functional excitatory clusters may be a biologically plausible way to implement weight sharing. Note, however, that in contrast to global weight sharing, this biological implementation is limited by the receptive field and the projective field of the interneurons, such that the generalisation will likely be spatially restricted. (iii) Stability of representations: Excitatory responses did not change during the rewarded phase. Therefore, the mechanism ensures a stable representation during relevant behaviour despite learning a structure between the inhibitory neurons.

Inhibitory connectivity is increasingly considered to be part of memory, e.g. refs. [52,53]. Mongillo et al.[53] show that a change in inhibitory connections has a larger impact on network activity than that of excitatory connections. This makes inhibitory synaptic changes well-suited to store memories, but also to trigger changes in excitatory connectivity. Here we propose that inhibitory connectivity stores a memory of the rewarded stimulus and can hence instruct changes in excitatory connectivity.

Interneuron circuits form canonical motifs across cortical areas. They integrate modulatory and long-range signals from higher cortical areas with activity in the local circuit. They are hence well-suited to adjust local circuits according to behaviourally relevant signals. We propose that interneuron circuits enable reward-dependent changes in sensory representations in a two-stage process. It can bridge timescales between stimulus-reward experience and synaptic plasticity. Finally, it allows for generalisation of the learned association.

## Methods

**Network model**. The network consisted of 400 PCs grouped into four subpopulations of 100 neurons each. Each subpopulation coded for a given orientation. We simulated 120 PV interneurons, 120 SST interneurons (30 in each subpopulation), and 50 VIP interneurons.

**Table 1 Parameters of the leaky integrate-and-fire neuron model**

| Parameter | Value | Parameter | Value |
|---|---|---|---|
| $C_m$ | 200 pF | $V_E$ | 0 mV |
| $V_l$ | −60 mV | $V_I$ | −80 mV |
| $g_l$ | 10 nS | $\tau_E$ | 5 ms |
| $v_\theta$ | −50 mV | $\tau_I$ | 10 ms |

*Neuron model.* Neurons were modelled as conductance-based spiking leaky integrate-and-fire neurons. Their membrane potential evolves according to:

$$C_m \frac{dv_i}{dt} = -g_l(v_i - V_l) - (g_i^E(v_i - V_E) + g_i^I(v_i - V_I)) + \sqrt{\frac{2\sigma^2}{\tau}}\xi(t) \qquad (1)$$

where $C_m$ is the membrane capacitance, $v_i$ is the membrane potential of neuron $i$, $V_l$ is the leak reversal potential. $V_E$ and $V_I$ are the excitatory and inhibitory reversal potentials. $g_l$, $g_i^E$ and $g_i^I$ are the leak, excitatory and inhibitory conductances. $g_i^E$ and $g_i^I$ are increased by $W_{ij}$ upon a spike event in a presynaptic excitatory or inhibitory neuron $j$, and decay exponentially with time constants $\tau_E$ and $\tau_I$, respectively:

$$\frac{dg_i^E}{dt} = -\frac{g_i^E}{\tau_E} + \sum_k W_{ij}\delta(t - t_j^k) \qquad (2)$$

$$\frac{dg_i^I}{dt} = -\frac{g_i^I}{\tau_I} + \sum_k W_{ij}\delta(t - t_j^k) \qquad (3)$$

$\xi$ is zero-mean Gaussian white noise. Parameters defining the Ornstein-Uhlenbeck process are $\sigma = 2$ mV and correlation time $\tau = 5$ ms.

When the membrane potential reaches a threshold $v_\theta$, a spike event is recorded and the membrane potential is reset to its resting value $V_l$ (Table 1). PVs were additionally connected via gap junctions which contribute a current $I_{gap,i}$ to the RHS of Eq. (1)[54]. The gap junction current from neuron $j$ to neuron $i$ is the sum of a spikelet current and a subthreshold current. The subthreshold current is proportional to the difference in membrane potential between neurons $i$ and $j$, with proportionality constant $w_{gap}$. The spikelet current is increased by $c_{gap}$ upon a spike event in a presynaptic neuron and decays to 0 with timescale $\tau_{spikelet}$. Mathematically,

$$I_{gap,i} = I_{spikelet,i} + \sum_j w_{gap}(v_j - v_i) \qquad (4)$$

$$\frac{dI_{spikelet,i}}{dt} = -\frac{I_{spikelet,i}}{\tau_{spikelet}} + \sum_k c_{gap}\delta(t - t_j^k) \qquad (5)$$

**Connectivity.** Neurons from different cell classes (E: PC, P: PV, S: SST, V:VIP) are connected with chemical synapses as follows:

The connection probability $P_{IJ}$ from population $J$ to $I$ where $I, J \in \{E, P, S, V\}$ was chosen based on data from Pfeffer et al.[21] for connections from inhibitory populations, and on data from Hofer et al.[20] for connections from the excitatory population. Pfeffer et al.[21] provide connection probabilities and strengths for connections between the pairs {EP,ES,EV,PP,PS} and individual neuronal contributions (INCs) estimated from optogenetic stimulation of entire cell populations for the other connections. In cases where the connection probability was not measured directly, we chose the probability based on the INCs as follows: For $P_{PV}$, $P_{VV}$, $P_{SS}$ and $P_{SP}$, the INC was very low, namely 0.06 (or 0.07 for $P_{SP}$). The connection from VIPs to PCs (EV) had the same INC of 0.06, therefore we chose the same connection probability as for $P_{EV}$, which was 12.5%. For the remaining connections, we set the connection probability to 100%.

$$P_{IJ} = \begin{pmatrix} P_{EE} & P_{EP} & P_{ES} & P_{EV} \\ P_{PE} & P_{PP} & P_{PS} & P_{PV} \\ P_{SE} & P_{SP} & P_{SS} & P_{SV} \\ P_{VE} & P_{VP} & P_{VS} & P_{VV} \end{pmatrix} = \begin{pmatrix} 1 & 1 & 1 & 0.125 \\ 0.88 & 1 & 0.857 & 0.125 \\ 1 & 0.125 & 0.125 & 1 \\ 1 & 1 & 1 & 0.125 \end{pmatrix} \qquad (6)$$

The synaptic weights $W_{IJ}$ from a neuron $j$ in population $J$ to a neuron $i$ in population $I$ determine how much the synaptic conductances $g^E$ and $g^I$ increase upon a spike in neuron $j$. We initialised the synaptic weights based on the connectivity data from Pfeffer et al.[21] for connections from inhibitory populations, and based on data from refs. [20,22–24] for connections from excitatory to inhibitory populations. The number of neurons in each population was taken into account, when determining the connection strength. Connections between excitatory neurons were initially small and sampled from a Gaussian distribution

truncated at 0.

$$W = \begin{pmatrix} W_{EE} & W_{EP} & W_{ES} & W_{EV} \\ W_{PE} & W_{PP} & W_{PS} & W_{PV} \\ W_{SE} & W_{SP} & W_{SS} & W_{SV} \\ W_{VE} & W_{VP} & W_{VS} & W_{VV} \end{pmatrix} = \begin{pmatrix} \mathcal{N}(0.01, 0.01) & 0.55 & 0.3 & 0.0675 \\ 0.12 & 0.55 & \mathcal{N}(0.2, 0.1) & 0.0675 \\ 0.07 & 0.08 & 0.0675 & 0.195 \\ 0.07 & 0.12 & 0.42 & 0 \end{pmatrix} \qquad (7)$$

For the gap junctions between PVs, $\tau_{spikelet}$ was 9 ms, except in Supplementary Fig. 4, where it was 3 ms. $c_{gap}$ was 13 pA, unless otherwise stated. Subthreshold currents mediated by gap junctions were modelled only in Supplementary Fig. 9, where $w_{gap}$ was 0.4 nS.

**Inputs.** PCs and SSTs received one of four inputs (corresponding to layer 4 (L4) inputs coding for four different orientations). Each L4 input produced a Poisson-distributed spike train with a rate of 4 kHz during its preferred stimulus, and 0 Hz otherwise. One of four stimuli was shown for 50 ms followed by a stimulus gap of 20 ms. During the stimulus gap, all L4 inputs produced spikes at the same rate of 1.6 kHz. The conductance of synapses from L4 to PCs was 0.28 nS. The conductances of synapses from L4 to SSTs was 0.15 nS during the stimulus and 0.165 nS during the stimulus gap. In addition, PCs and PVs received a baseline input from a Poisson process with a rate of 4 kHz. The weights to PCs were 0.13 nS, and to PVs 0.01 nS. VIPs received a top-down input during the vertical bar stimulus from a group of 100 neurons with connection strength 0.2 nS, which receive input from layer 4 with a connection strength of 0.3 nS.

**Plasticity.** For both excitatory and inhibitory plasticity we chose the simple classical STDP model[55–57]

$$\Delta w = \begin{cases} -A_- \exp(\frac{\Delta t}{\tau_-}) & \text{if } \Delta t < 0 \\ A_+ \exp(-\frac{\Delta t}{\tau_+}) & \text{if } \Delta t \geq 0 \end{cases} \qquad (8)$$

where $\Delta t = t_{post} - t_{pre}$ is the difference between pre- and postsynaptic spike time, $\tau_+ = \tau_- = 20$ ms, for excitatory plasticity $A_+ = 0.005$ nS and $A_- = 1.05 A_+$, for inhibitory plasticity $A_+ = 0.015$ nS and $A_- = 1.05 A_+$. This rule leads to synaptic potentiation when the presynaptic neuron spikes before or simultaneously with the postsynaptic neuron, and to depression otherwise.

In the online implementation of this rule, the synaptic weight $w_{ij}$ from neuron $j$ to neuron $i$ is updated when either the pre- or the postsynaptic neuron spikes according to:

$$w_{ij} \rightarrow w_{ij} - \eta a_{post}(t) \qquad \text{for presynaptic spikes at time } t \qquad (9)$$

$$w_{ij} \rightarrow w_{ij} + \eta a_{pre}(t) \qquad \text{for postsynaptic spikes at time } t \qquad (10)$$

where $a_{post}(t)$ is the postsynaptic trace and $a_{pre}(t)$ is the presynaptic trace. The traces are updated by a constant value $A_-$ or $A_+$ at the time of a postsynaptic or presynaptic spike, respectively, and decay exponentially with a time constant $\tau_-$ or $\tau_+$. The learning rate $\eta$ was 1.

$$\frac{da_{post}}{dt} = -\frac{a_{post}}{\tau_-} \qquad (11)$$

$$\frac{da_{pre}}{dt} = -\frac{a_{pre}}{\tau_+} \qquad (12)$$

Excitatory and inhibitory weights were constrained to be positive and had an upper bound at 0.25 nS for excitatory weights and 1 nS for inhibitory weights.

**Simulation.** All spiking simulations were done with the Brian 2 simulator[58], using a time step of 0.1 ms. The model was simulated for 1.4 s without plasticity to measure tuning curves. Then plasticity was switched on. The model was simulated for 42 s during the developmental phase, followed by a 24.5 s simulation of the rewarded phase, and a 66 s simulation of the refinement phase. Finally, the model was simulated for 1.4 s without plasticity to measure final tuning curves again.

**Translation invariance.** To adjust for the increased number of excitatory neurons in the network, the connection strength from PCs to all interneurons was decreased to 0.6 times the original strength.

**Reporting summary.** Further information on research design is available in the Nature Research Reporting Summary linked to this article.

## Data availability

All data is generated by the simulation code (see Code availability statement below).

## Code availability

All simulation code used for this paper is available on GitHub (https://github.com/k47h4/interneuron_circuits_plasticity) and ModelDB (accession number 259546).

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

## Acknowledgements

This work was supported by DFG 398005926, BBSRC BB/N013956/1, BB/N019008/1, Wellcome Trust 200790/Z/16/Z, Simons Foundation 564408, EPSRC EP/R035806/1. We

thank Friedemann Zenke and Paul Chadderton for comments on the paper and Yann Sweeney for helpful discussions.

## Author contributions

K.W. and C.C. conceived the project and designed the experiments. K.W. performed the simulations and analysed the data. K.W and C.C. interpreted the results and wrote the paper.

## Competing interests

The authors declare no competing interests.
