## [Peer Review File · Nature Communications]

Reviewers' Comments:

Reviewer #1:

Remarks to the Author:

This paper proposes a model for reward-driven plasticity of neural responses in visual cortex. The mechanism involves plasticity of inhibitory neuron inter-connectivity, which disinhibits pyramidal neurons, whose interconnections can then themselves exhibit plasticity. The mechanism is demonstrated using simulations and predictions of the model are discussed.

While it is brief, the paper describes a clear model and has testable predictions. It proposes a novel role for inhibitory cortical circuitry and is thus of contemporary interest. I do feel that the authors could do a better job of discussing the model assumptions and limitations, and of placing these in the context of existing literature. I also found some claims about translation invariance to be strong.

Major comments:

Last I checked, the extent to which different subtypes of inhibitory neurons are tuned or untuned is still a matter of debate (e.g. the work of Clay Reid, Mriganka Sur). Considering that the model makes strong assumptions on this, the authors should discuss in more detail this literature.

The section on the translation invariance of learned representations seems somewhat farfetched. Wouldn't the proposed mechanism only work within the distance of the spatial receptive field of inhibition? Even if this is broader than excitation, it is still far from the "weight sharing" of convolutional networks as mentioned in the discussion. How broadly tuned are inhibitory neurons compared to excitatory neurons?

A recent paper from Mongillo, Rumpel & Loewenstein (Nature Neurosci. 2018) also has a related hypothesis about the interplay between excitatory and inhibitory plasticity. The relationship to this work should be discussed in the paper.

Minor comments:

Abstract: "increased stimulus representation" -- be more specific here.

Final sentence of the discussion: This is also a vague statement and should be made more precise.

Reviewer #2:

Remarks to the Author:

The manuscript addresses an important and timely question: what is the contribution of inhibitory circuits in the formation of stimulus/reward associations. The authors propose a theoretical framework based on a network model that includes three different populations of GABAergic neurons (VIP, SST and PV) connected according to reports from experimental work. In the model, VIP neurons are excited by the reward associated to a stimulus with a specific orientation, thus inhibiting SST neurons postsynaptic to VIP which are tuned for the rewarded stimulus. SST neurons are presynaptic to PV (untuned in this model). In principle, this process would disinhibit PV neurons in response to the rewarded orientation, while PV neurons should still be inhibited by the non-rewarded orientations. However, the authors propose that an STDP-like plasticity is induced upon multiple pairings of the stimulus/reward, leading to increased connectivity of SST and PV, and consequent disinhibition of pyramidal neurons. Finally, in the second phase (the refinement phase), the modifications in inhibitory connectivity result in facilitation of glutamatergic plasticity between pyramidal neurons tuned to the rewarded orientation.

While in my opinion this type of theoretical work is important, innovative and can potentially provide interesting predictions about circuit mechanisms for stimulus/reward associations, the logic of the theory presented in this study is unclear.

Major concerns:

1- How would a standard STDP rule induce potentiation at the SST-PV connection? STDP is based on the idea that if presynaptic firing precedes postsynaptic firing a connection is strengthened, while the same connection is weakened if postsynaptic activity precedes presynaptic firing. In the model scheme proposed here, VIP neurons inhibit SST neurons, while PV neurons become disinhibited. Unless some other factor were involved, low activity in SST neurons would be paired with PV neurons being disinhibited by the rewarded stimulus and inhibited by the others. Intuitively, this suggests a post before pre pairing, which by standard STDP rules would result in depression, not potentiation. What is the basis for expecting SST-PV potentiation?

2- Why do the authors propose a 2-step process? What is the rationale for not having all the plasticity interactions occur simultaneously or nearly simultaneously to the arrival of the reward?

3- In principle, a network could learn the stimulus/reward association by weakening connections to non-rewarded stimuli. Have the authors considered this possibility?

4- The connectivity between neuron groups in the model is all-to-all. As the authors' goal is to use experimental data to guide their theoretical framework, why wasn't the differential connectivity between neuron groups implemented in the model? Pfeiffer et al, 2013, which the authors cite, provides information regarding connection probability within and between neuron groups. Connection probability between pyramidal neurons as well as between inhibitory neurons and pyramidal neurons is also widely reported. Differential connectivity between neuron groups could also expand the parameter space for the stimulus/ reward association.

Minor concerns:

Some of the terminology used in the manuscript is unclear.

1- In the introduction the authors state: "We hypothesized that the inhibitory circuit in L2/3 mediates the top down instructions to guide slow plastic changes...." What would the "slow plastic changes" be?

2- At the end of the introduction the authors state: "Our model offers testable predictions on the activity of different cell types during and after reward presentation". This is actually not what the model does. The model makes predictions about functional interactions between neuron groups, but does not provide any prediction in regard of what the actual activity of the different cell types.

3- In the results: "Activity dependent plasticity then increases the connections between SST which are tuned to the rewarded stimulus and PV". What do the authors mean with "increases connections"? Would this be increase in number of connections, or increased functional connectivity, or increased synaptic strength?

Arianna Maffei

Reviewer #3:

Remarks to the Author:

In this manuscript, Wilmes and Clopath propose a model of V1 microcircuitry that produces reward related sensory plasticity. The proposed model produces interesting dynamics during and after learning. It also takes into account the three main interneuron types observed in V1, which is commendable. However, I have major concerns with this manuscript. The most important one is that the interesting aspect of this model (i.e. two-stage learning) is highly counterintuitive and is contradicted by available experimental data (# 1 below). The remaining aspect of the model (SST to PV plasticity) is what speaks to the bulk of experimental data showing reward-related sensory plasticity. However, the model's explanation for this literature is just obtained by design due to some restrictive assumptions. On a related note, assumptions of the model are often not explicitly laid out.

Indeed, some crucial assumptions are not experimentally supported, either due to lack of evidence or evidence to the contrary. Overall, I think the manuscript needs significant work to advance the field.

1. A key assumption/prediction of the model is that stimulus selectivity in PCs is increased during the refinement phase in addition to the rewarded phase. In more behaviorist terms, even after extinction of a CS-US association, stimulus selectivity for a previously rewarded CS will increase. In addition to being a highly counterintuitive falsifiable prediction for primary sensory cortices, available data directly contradict this (Bieszczad and Weinberger, Extinction reveals that primary sensory cortex predicts reinforcement outcome, *Eur. J. Neurosci.*, 2012). A related falsifiable prediction that is not explicitly laid out is that recurrent connections between excitatory neurons do not strengthen during the rewarded phase (Fig 2f). This predicts that sensory plasticity in the presence of reward (i.e. almost all related experimental data) occurs without any changes in recurrent connections. This aspect has not been directly tested as far as I know, but seems quite counterintuitive. Also, I suspect that relaxing assumptions to be more in line with experimental data (#2 - #6) will have a significant impact on these findings. Thus, I highly doubt that this feature of the model will survive experimental tests.

Once this aspect is removed, all that the current model says is that SST to PV synaptic potentiation causes reward-related sensory plasticity. In my opinion, this result does not advance the field much as it is obtained by design in this model: the authors assume that only SST to PV synapse is plastic in addition to recurrent synapses. More importantly, this one-stage model is the model that addresses the bulk of experimental data as most experimental data only relate to the "rewarded phase" of this work. Overall, I find that for established experimental findings, the model presented here is trivial. And the interesting aspects of the model are either experimentally invalidated in case related data are available, or highly counterintuitive and unlikely to be verified.

2. It is assumed that only connections from SSTs to PVs and recurrent connections between PCs show synaptic plasticity. SST to PC or VIP to SST or PV to PC strengths are assumed to be fixed. What evidence is there to support these assumptions? Most importantly, what evidence is there to assume that SST to PV synapses show plasticity but not SST to PC synapses? My guess is that plasticity at SST to PC synapses will affect the refinement process mentioned here. Further, PV to PC synapses have been shown to undergo STDP (Vickers et al., Parvalbumin-Interneuron Output Synapses Show Spike-Timing-Dependent Plasticity that Contributes to Auditory Map Remodeling, *Neuron*, 2018). How does this affect the model, especially considering that PV cells may also exhibit stimulus selectivity?

3. The authors assume that PV neurons show broad stimulus-untuned responses. While some literature does exist supporting this assumption (as the authors cite), a lot of papers show spatial tuning for PV neurons. For example, Cardin et al. show that fast-spiking interneurons exhibit clear stimulus-selectivity (Cardin et al., Stimulus Feature Selectivity in Excitatory and Inhibitory Neurons in Primary Visual Cortex, *J. Neurosci.* 2007); numerous papers show that most parvalbumin neurons are fast-spiking (e.g. Hu et al. Interneurons. Fast-spiking, parvalbumin⁺ GABAergic interneurons: from cellular design to microcircuit function, *Science*, 2014). Additional papers also support spatial tuning in genetically identified PV neurons (Runyan et al., Response features of parvalbumin-expressing interneurons suggest precise roles for subtypes of inhibition in visual cortex, *Neuron* 2010; Wilson et al., Division and subtraction by distinct cortical inhibitory networks in vivo, *Nature* 2012). Thus, this fundamental aspect of their model seems to be at odds with considerable experimental data.

4. SSTs inhibit VIPs equally strongly as they inhibit PCs (Pfeffer et al. 2013). This connection is not included in the model. How do results depend on this connection?

5. PV to PC connection strength is approximately twice that of SST to PC strength (Pfeffer et al. 2013). In this model, the strengths are assumed to be 0.25 and 0.3, respectively, which is at odds with experimental data. My guess is that increasing the PV to PC strength to reflect experimental data will reduce the excitatory structure index during refinement.

6. The interesting aspects of the current model primarily work via disinhibition of PCs by SSTs. However, data show that the primary effect of SSTs on PCs is inhibition, not disinhibition (e.g. Lee et al. Activation of specific interneurons improves V1 feature selectivity and visual perception, *Nature*

2012). Is the net effect of SSTs on PCs inhibition or disinhibition in the model, especially after #5 above is addressed?

7. How do the authors account for a reduction in stimulus selectivity for unrewarded stimuli as was observed by Poort et al. 2015?

8. The authors may want to cite Kilgard and Merzenich, Cortical map reorganization enabled by nucleus basalis activity, *Science*, 1998.

9. Reward is not a behaviorally relevant "context". It is a stimulus.

10. "We wanted to test whether interneurons can learn from a top-down signal." This statement implies that an experimental test is conducted in the present study and is thus misleading. It should instead say something like, "Based on our current experimental understanding, we wanted to test whether interneurons can learn from a top-down signal in a model of cortical circuitry."

Reviewer #1:

Last I checked, the extent to which different subtypes of inhibitory neurons are tuned or untuned is still a matter of debate (e.g. the work of Clay Reid, Mriganka Sur). Considering that the model makes strong assumptions on this, the authors should discuss in more detail this literature.

Good point! We will discuss the tuning of interneurons in more detail.

The section on the translation invariance of learned representations seems somewhat farfetched. Wouldn't the proposed mechanism only work within the distance of the spatial receptive field of inhibition? Even if this is broader than excitation, it is still far from the "weight sharing" of convolutional networks as mentioned in the discussion. How broadly tuned are inhibitory neurons compared to excitatory neurons?

We will test in our model the ranges of tuning in agreement with the proposed mechanism.

A recent paper from Mongillo, Rumpel & Loewenstein (Nature Neurosci. 2018) also has a related hypothesis about the interplay between excitatory and inhibitory plasticity. The relationship to this work should be discussed in the paper.

Great comment! We will discuss our work in the light of Mongillo et al.'s work. This work shows that a change in inhibitory connections has larger impact on network activity than that of excitatory connections. They conclude that therefore, memories should ultimately be stored in inhibitory connections. This is related to our work as we also propose a role of inhibitory connections in storage, albeit in a different sense. Additionally, we think that this property of inhibitory synaptic changes makes them also very well suited to trigger changes in excitatory connectivity.

We will also test the impact of synaptic turnover in our model.

Reviewer #2:

The manuscript addresses an important and timely question: what is the contribution of inhibitory circuits in the formation of stimulus/reward associations. The authors propose a theoretical framework based on a network model that includes three different populations of GABAergic neurons (VIP, SST and PV) connected according to reports from experimental work. In the model, VIP neurons are excited by the reward associated to a stimulus with a specific orientation, thus inhibiting SST neurons postsynaptic to VIP which are tuned for the rewarded stimulus. SST neurons are presynaptic to PV (untuned in this model). In principle, this process would disinhibit PV neurons in response to the rewarded orientation, while PV neurons should still be inhibited by the non-rewarded orientations. However, the authors propose that an STDP-like plasticity is induced upon multiple pairings of the stimulus/reward, leading to increased connectivity of SST and PV, and consequent disinhibition of pyramidal neurons. Finally, in the second phase (the refinement phase), the modifications in inhibitory connectivity result in facilitation of glutamatergic plasticity between pyramidal neurons tuned to the rewarded orientation. While in my opinion this type of theoretical work is important, innovative and can potentially provide interesting predictions about circuit mechanisms for stimulus/reward associations, the logic of the theory presented in this study is unclear.

Major concerns:

1- How would a standard STDP rule induce potentiation at the SST-PV connection? STDP is based on the idea that if presynaptic firing precedes postsynaptic firing a connection is strengthened, while the same connection is weakened if postsynaptic activity precedes presynaptic firing. In the model scheme proposed here, VIP neurons inhibit SST neurons, while PV neurons become disinhibited. Unless some other factor were involved, low activity in SST neurons would be paired with PV neurons being disinhibited by the rewarded stimulus and inhibited by the others. Intuitively, this suggests a post before pre pairing, which by standard STDP rules would result in depression, not potentiation. What is the basis for expecting SST-PV potentiation?

SSTs are highly responsive to visual stimuli and quickly recruited by thalamic stimulation. Therefore, SSTs could fire before they are suppressed by VIPs. Optogenetic inhibition of SSTs (mimicking inhibition by VIPs) increases the firing rate of L4 FS interneurons (aka PVs) subsequently (Xu et al. "Neocortical Somatostatin-Expressing GABAergic Interneurons Disinhibit the Thalamorecipient Layer 4", Neuron, 2013), such that SSTs fire before PVs. Classical STDP, which strengthens the pre-before-post firing, therefore increases the SST-to-PV connection.

2- Why do the authors propose a 2-step process? What is the rationale for not having all the plasticity interactions occur simultaneously or nearly simultaneously to the arrival of the reward?

We described the mechanism in 2 steps to clarify the different parts of the process, not because the 2 steps need to happen separately. Indeed, both excitatory and inhibitory network structures can develop at the same time (see supplementary figures S4 and S5). We presented a parameter setting in the main manuscript in which the excitatory structure develops later for three reasons: First, separating the two stages logically makes them easier to understand. Second, it becomes immediately clear that the inhibitory structure alone can induce the excitatory changes. If the excitatory network developed at the same time, it would not be clear whether the further development is due to a positive feedback loop in the excitatory network or due to the inhibitory structure. Third, we consider it a feature that the two structures can develop separately.

3- In principle, a network could learn the stimulus/reward association by weakening connections to non-rewarded stimuli. Have the authors considered this possibility?

Very good point, we thought about this possibility. To selectively increase the plasticity during the rewarded stimulus (the goal of the inhibitory structure), the inhibitory structure has to disinhibit the PCs during the rewarded stimulus. The suggested weakening of connections from other SSTs to PVs would decrease the amount of disinhibition during unrewarded stimuli. Although, disinhibition during the rewarded stimulus would be increased relative to the others, this does not enhance disinhibition during

the rewarded stimulus relative to baseline. PCs would not fire more together during the rewarded stimulus (the goal of the structure), but rather even less during unrewarded stimuli.

4- The connectivity between neuron groups in the model is all-to-all. As the authors' goal is to use experimental data to guide their theoretical framework, why wasn't the differential connectivity between neuron groups implemented in the model? Pfeffer et al, 2013, which the authors cite, provides information regarding connection probability within and between neuron groups. Connection probability between pyramidal neurons as well as between inhibitory neurons and pyramidal neurons is also widely reported. Differential connectivity between neuron groups could also expand the parameter space for the stimulus/ reward association.

Good point! We will address this. As mentioned earlier, we are confident that the more realistic connectivity will not compromise the proposed mechanism.

Minor concerns:

Some of the terminology used in the manuscript is unclear.

1- In the introduction the authors state: "We hypothesized that the inhibitory circuit in L2/3 mediates the top down instructions to guide slow plastic changes...." What would the "slow plastic changes" be?

We were referring to changes in the excitatory connectivity. We referred to them as slow as they may require more time than available during the rewarded phase. We will rephrase this sentence.

2- At the end of the introduction the authors state: "Our model offers testable predictions on the activity of different cell types during and after reward presentation". This is actually not what the model does. The model makes predictions about functional interactions between neuron groups, but does not provide any prediction in regard of what the actual activity of the different cell types.

The model makes predictions about firing rates for each of the cell types. We provided mean tuning curves for PCs, PVs and SSTs, and firing rate changes for PCs. We will include firing rate distributions in the supplementary material for all cell types, including VIPs.

3- In the results: "Activity dependent plasticity then increases the connections between SST which are tuned to the rewarded stimulus and PV". What do the authors mean with "increases connections"? Would this be increase in number of connections, or increased functional connectivity, or increased synaptic strength?

The synaptic strength increased.

Reviewer #3:

In this manuscript, Wilmes and Clopath propose a model of V1 microcircuitry that produces reward related sensory plasticity. The proposed model produces interesting dynamics during and after learning. It also takes into account the three main interneuron types observed in V1, which is commendable. However, I have major concerns with this manuscript. The most important one is that the interesting aspect of this model (i.e. two-stage learning) is highly counterintuitive and is contradicted by available experimental data (# 1 below). The remaining aspect of the model (SST to PV plasticity) is what speaks to the bulk of experimental data showing reward-related sensory plasticity. However, the model's explanation for this literature is just obtained by design due to some restrictive assumptions. On a related note, assumptions of the model are often not explicitly laid out. Indeed, some crucial assumptions are not experimentally supported, either due to lack of evidence or evidence to the contrary. Overall, I think the manuscript needs significant work to advance the field.

We agree that we should give more details on the model assumptions. We respond to the remaining points in more detail below.

1. A key assumption/prediction of the model is that stimulus selectivity in PCs is increased during the refinement phase in addition to the rewarded phase. In more behaviorist terms, even after extinction of a CS-US association, stimulus selectivity for a previously rewarded CS will increase. In addition to being a highly counterintuitive falsifiable prediction for primary sensory cortices, available data directly contradict this (Bieszczad and Weinberger, Extinction reveals that primary sensory cortex predicts reinforcement outcome, Eur. J. Neurosci., 2012). A related falsifiable prediction that is not explicitly laid out is that recurrent connections between excitatory neurons do not strengthen during the rewarded phase (Fig 2f). This predicts that sensory plasticity in the presence of reward (i.e. almost all related experimental data) occurs without any changes in recurrent connections. This aspect has not been directly tested as far as I know, but seems quite counterintuitive.

Extinction is a very interesting phenomenon that we did not model in our study. We would like to clarify that our model does not make predictions for what happens after extinction of a CS-US association. The cited study by Bieszczad and Weinberger shows that rats have a similar sized area representing a previously rewarded frequency after extinction (occurring after 3 days) compared to control. On the contrary, animals that learned an association between a frequency and a reward show an increased representation of that frequency. Hence, the change in representation seems to vanish with extinction. This study however does not reveal whether the original synaptic changes are being reversed. A study in the amygdala (Grewe et al. "Neural ensemble dynamics underlying a long-term associative memory", Nature, 2017) shows that the change in representation with extinction is not a reversal of the changes caused by association learning. Indeed, this study makes another interesting point: it shows that the changes in CS+ representation mostly happen in a consolidation phase a day after learning as opposed to during learning. Therefore, the presentation of the CS+ alone does not immediately lead to extinction, but on the contrary changes are still reinforced. Therefore, although counterintuitive, the experimental data

seems to point to a similar direction as our model. While it is undoubtedly an interesting question of what the neural correlate of extinction is, it is beyond the scope of our study. Extinction was not reported by Poort, Khan et al. ("Learning Enhances Sensory and Multiple Non-sensory Representations in Primary Visual Cortex", Neuron, 2015), when they alternated between visual and olfactory blocks. In the olfactory blocks, the visual stimuli were uninformative about reward. Hence, the vertical grating was not always paired with a reward.

The second point is related to point 2 of reviewer 2. We will clarify that recurrent connections can change in the presence of the reward.

Once this aspect is removed, all that the current model says is that SST to PV synaptic potentiation causes reward-related sensory plasticity. In my opinion, this result does not advance the field much as it is obtained by design in this model: the authors assume that only SST to PV synapse is plastic in addition to recurrent synapses. More importantly, this one-stage model is the model that addresses the bulk of experimental data as most experimental data only relate to the "rewarded phase" of this work. Overall, I find that for established experimental findings, the model presented here is trivial. And the interesting aspects of the model are either experimentally invalidated in case related data are available, or highly counterintuitive and unlikely to be verified.

2. It is assumed that only connections from SSTs to PVs and recurrent connections between PCs show synaptic plasticity. SST to PC or VIP to SST or PV to PC strengths are assumed to be fixed. What evidence is there to support these assumptions? Most importantly, what evidence is there to assume that SST to PV synapses show plasticity but not SST to PC synapses? My guess is that plasticity at SST to PC synapses will affect the refinement process mentioned here. Further, PV to PC synapses have been shown to undergo STDP (Vickers et al., Parvalbumin-Interneuron Output Synapses Show Spike-Timing-Dependent Plasticity that Contributes to Auditory Map Remodeling, Neuron, 2018). How does this affect the model, especially considering that PV cells may also exhibit stimulus selectivity?

We will address these questions by adding plasticity to the other connections in the model.

STDP at SST-to-PC connections will lead to an increase in the connection from vertically-tuned SSTs to PCs during the rewarded phase (as was the case between SSTs and PVs). As the SSTs are suppressed during the rewarded phase, this will not have a large effect on network activity. During the refinement phase, the SSTs will still disinhibit the PCs via PV inhibition more strongly during the vertical bar than during unrewarded stimuli and hence the selective disinhibition in the network (the relevant mechanism) will remain. Taking into account that SSTs target the apical dendrites, the increased SST-to-PC inhibition will affect the distal apical dendrites of pyramidal cells, and hence the main mechanism during the refinement phase, the somatic disinhibition of PCs via inhibition of PVs, will remain.

Anti-Hebbian PV-to-PC plasticity will strengthen connections from PVs to PCs, which are driven by PCs and hence follow the PCs in firing. The stronger the PV-to-PC inhibition, the stronger the disinhibitory effect of the SSTs, which is beneficial for the proposed mechanism. If PVs are stimulus-selective, and even if PVs and

PCs with similar responses cluster together, the mechanism still works. It only requires that connections from SSTs to PVs are not response-specific. Then SSTs will suppress PVs of all selectivities and disinhibit PCs of all selectivities.

3. The authors assume that PV neurons show broad stimulus-untuned responses. While some literature does exist supporting this assumption (as the authors cite), a lot of papers show spatial tuning for PV neurons. For example, Cardin et al. show that fast-spiking interneurons exhibit clear stimulus-selectivity (Cardin et al., Stimulus Feature Selectivity in Excitatory and Inhibitory Neurons in Primary Visual Cortex, J. Neurosci. 2007); numerous papers show that most parvalbumin neurons are fast-spiking (e.g. Hu et al. Interneurons. Fast-spiking, parvalbumin⁺ GABAergic interneurons: from cellular design to microcircuit function, Science, 2014). Additional papers also support spatial tuning in genetically identified PV neurons (Runyan et al., Response features of parvalbumin-expressing interneurons suggest precise roles for subtypes of inhibition in visual cortex, Neuron 2010; Wilson et al., Division and subtraction by distinct cortical inhibitory networks in vivo, Nature 2012).

Thus, this fundamental aspect of their model seems to be at odds with considerable experimental data.

We do not make assumptions about the spatial tuning of PVs, as this does not influence the mechanism that we propose. We cited studies showing that PVs are broadly tuned to orientation, not space. There are several studies that support this broad tuning, that we did not cite: e.g. Sohya K et al. "GABAergic neurons are less selective to stimulus orientation than excitatory neurons in layer 2/3 of visual cortex", Journal of Neuroscience 2007.

Moreover, the study by Cardin et al. (mentioned by the reviewer) shows that FS cells are only a bit less tuned than regular spiking (RS) cells in layer 4, but these are cells in layer 4 and not in layer 2/3. This study also shows that RS cells in layer 4 are more sharply tuned than in layer 2/3, which are in turn more sharply tuned than fast spiking (FS) cells in layer 2/3. The general opinion seems to be that interneurons are less sharply tuned to orientation than excitatory cells.

4. SSTs inhibit VIPs equally strongly as they inhibit PCs (Pfeffer et al. 2013). This connection is not included in the model. How do results depend on this connection?

We will include this connection into the model as we are going to use more realistic connectivity. We only introduced a simple unidirectional connection to model the experimental observation that VIPs potently inhibit SSTs in several cortical areas (e.g. Williams et al. "Higher-Order Thalamocortical Inputs Gate Synaptic Long-Term Potentiation via Disinhibition", Neuron 2019). In a recent theoretical study (Hertaeg et al "Amplifying the redistribution of somato-dendritic inhibition by the interplay of three interneuron types", bioRxiv 2018), which investigated the difference between unidirectional and bidirectional connections between VIPs and SSTs, it was shown that also with bidirectional connections, VIPs can silence SSTs. Therefore, we don't expect our model with realistic connectivity to yield different results. We will add new simulations in the revision.

5. PV to PC connection strength is approximately twice that of SST to PC strength (Pfeffer et al. 2013). In this model, the strengths are assumed to be 0.25 and 0.3, respectively, which is at odds with experimental data. My guess is that increasing the PV to PC strength to reflect experimental data will reduce the excitatory structure index during refinement.

To the contrary, it will increase the excitatory structure index, because the connectivity change will increase the disinhibition of PCs during the rewarded stimulus in the refinement phase. We will add new simulations highlighting that.

6. The interesting aspects of the current model primarily work via disinhibition of PCs by SSTs. However, data show that the primary effect of SSTs on PCs is inhibition, not disinhibition (e.g. Lee et al. Activation of specific interneurons improves V1 feature selectivity and visual perception, Nature 2012). Is the net effect of SSTs on PCs inhibition or disinhibition in the model, especially after #5 above is addressed?

During the rewarded phase, the PCs are disinhibited via VIP-SST during the rewarded stimulus. In the refinement phase, they are disinhibited via SST-PV during the rewarded stimulus. The disinhibitory effect of SSTs on PCs is learned during the rewarded phase and stimulus-specific. Their net effect is inhibition, not disinhibition. We will provide additional simulations to show that.

7. How do the authors account for a reduction in stimulus selectivity for unrewarded stimuli as was observed by Poort et al. 2015?

In Poort et al. 2015, the selectivity to both stimuli increased.

8. The authors may want to cite Kilgard and Merzenich, Cortical map reorganization enabled by nucleus basalis activity, Science, 1998.

Good idea!

9. Reward is not a behaviorally relevant “context”. It is a stimulus.

We agree that ‘context’ is an ill-defined term. We will be more precise in our wording. We treat reward as a stimulus in our model and will refer to it as a stimulus.

10. “We wanted to test whether interneurons can learn from a top-down signal.” This statement implies that an experimental test is conducted in the present study and is thus misleading. It should instead say something like, “Based on our current experimental understanding, we wanted to test whether interneurons can learn from a top-down signal in a model of cortical circuitry.”

Point-by-point rebuttal to the reviewer comments:

We are very grateful for the time and effort that the reviewers spent to give us valuable feedback on our manuscript. We hope that we have fully addressed their concerns. To that end, we added 5 new figures, several explanations in the main text and 2 tables (S1 and S2) to make the model's assumptions and predictions explicit.

Reviewer #1:

Last I checked, the extent to which different subtypes of inhibitory neurons are tuned or untuned is still a matter of debate (e.g. the work of Clay Reid, Mriganka Sur). Considering that the model makes strong assumptions on this, the authors should discuss in more detail this literature.

We agree that interneuron tuning is still a matter of debate. As suggested, we added a more detailed literature review of the tuning of interneurons to the discussion section of the manuscript ('Orientation tuning of interneurons'). We also simulated the network with tuned PV cells, and show that our results are robust with regard to the tuning of PVs (see Suppl. Fig. S5).

The section on the translation invariance of learned representations seems somewhat farfetched. Wouldn't the proposed mechanism only work within the distance of the spatial receptive field of inhibition? Even if this is broader than excitation, it is still far from the "weight sharing" of convolutional networks as mentioned in the discussion. How broadly tuned are inhibitory neurons compared to excitatory neurons?

This is a very good point! The reviewer is absolutely right in that the extent of the generalisation is limited by the spatial tuning width of the interneurons. We now state this clearly in the results section and toned down the claim in the discussion section.

Second question about inhibitory vs excitatory tuning: There are not many studies on the spatial tuning of interneurons (we cited Liu et al. 2009 for broader spatial tuning of interneurons than PCs). In our model however, a broader tuning of interneurons, albeit being small, can lead to the generalisation of the learned representation to nearby receptive field locations.

A recent paper from Mongillo, Rumpel & Loewenstein (Nature Neurosci. 2018) also has a related hypothesis about the interplay between excitatory and inhibitory plasticity. The relationship to this work should be discussed in the paper.

That's a very good point. We added a section to the discussion, where we now discuss our work in the light of Mongillo et al.'s work (see discussion section 'Inhibitory engrams'). Mongillo et al. show that a change in inhibitory connections has a larger impact on network activity than that of excitatory connections. They conclude that therefore, memories should ultimately be stored in inhibitory connections. This is related to our work as we also propose a role of inhibitory connections in learning, but here the key role is maintaining selective disinhibition for learning (e.g. Letzkus et al. 2011, Williams and Holtmaat, 2019, Adler et al. 2019).

Reviewer #2:

The manuscript addresses an important and timely question: what is the contribution of inhibitory circuits in the formation of stimulus/reward associations. The authors propose a theoretical framework based on a network model that includes three different populations of GABAergic neurons (VIP, SST and PV) connected according to reports from experimental work. In the model, VIP neurons are excited by the reward associated to a stimulus with a specific orientation, thus inhibiting SST neurons postsynaptic to VIP which are tuned for the rewarded stimulus. SST neurons are presynaptic to PV (untuned in this model). In principle, this process would disinhibit PV neurons in response to the rewarded orientation, while PV neurons should still be inhibited by the non-rewarded orientations. However, the authors propose that an STDP-like plasticity is induced upon multiple pairings of the stimulus/reward, leading to increased connectivity of SST and PV, and consequent disinhibition of pyramidal neurons. Finally, in the second phase (the refinement phase), the modifications in inhibitory connectivity result in facilitation of glutamatergic plasticity between pyramidal neurons tuned to the rewarded orientation. While in my opinion this type of theoretical work is important, innovative and can potentially provide interesting predictions about circuit mechanisms for stimulus/reward associations, the logic of the theory presented in this study is unclear.

Major concerns:

1- How would a standard STDP rule induce potentiation at the SST-PV connection? STDP is based on the idea that if presynaptic firing precedes postsynaptic firing a connection is strengthened, while the same connection is weakened if postsynaptic activity precedes presynaptic firing. In the model scheme proposed here, VIP neurons inhibit SST neurons, while PV neurons become disinhibited. Unless some other factor were involved, low activity in SST neurons would be paired with PV neurons being disinhibited by the rewarded stimulus and inhibited by the others. Intuitively, this suggests a post before pre pairing, which by standard STDP rules would result in depression, not potentiation. What is the basis for expecting SST-PV potentiation?

We apologise for the lack of clarity. In our model, SSTs are highly responsive to visual stimuli as they are quickly recruited by local excitatory and potentially by thalamic inputs (Pakan et al. “Behavioral-state modulation of inhibition is context-dependent and cell type specific in mouse visual cortex”, eLife, 2016). Therefore, SSTs fire before they are suppressed by VIPs (see Figure 1 below). A suppression of SST activity after they started firing then increases PV firing subsequently. This is similar to experimental optogenetic inhibition of SSTs (mimicking inhibition by VIPs) increasing the firing rate of L4 FS interneurons (aka PVs) subsequently (Xu et al. “Neocortical Somatostatin-Expressing GABAergic Interneurons Disinhibit the Thalamorecipient Layer 4”, Neuron, 2013). Hence, SSTs fire before PVs (see cross-correlogram in Figure 1 below). Classical STDP, which strengthens the pre-before-post firing, therefore increases the SST-to-PV connection.

Figure 1 Left: Cross-correlogram between a vertically-tuned SST cell and a PV cell. Time lag 0 is the time of a spike in the PV cell. The SST cell hence tends to fire before the PV cell. Right: Spike raster plot during rewarded phase. Vertically-tuned SSTs (bottom, purple) fire briefly before they are suppressed. PVs increase their firing rate after the SSTs fired (Figure taken from Suppl. Fig. S1).

2- Why do the authors propose a 2-step process? What is the rationale for not having all the plasticity interactions occur simultaneously or nearly simultaneously to the arrival of the reward?

We described the mechanism in two steps to clarify the different parts of the process, not because the two steps need to happen separately. Indeed, both excitatory and inhibitory network structures can develop at the same time (see Suppl. Fig. S3). We presented a parameter setting in the main manuscript in which the excitatory structure develops later for three reasons: First, for didactic reasons, it is easier to understand the two stages separately. Second, it becomes immediately clear that the inhibitory structure alone can induce the excitatory changes. If the excitatory network developed at the same time, it would not be clear whether the further development is due to a positive feedback loop in the excitatory network or due to the inhibitory structure. Third, we consider it a feature that the two structures can develop separately. If a

downstream area reads out from excitatory neurons only, this read-out would not change during the task, such that a separate development allows for a more stable representation/coding during the behaviour by postponing the change in representation to subsequent periods of rest. We made this more explicit in the manuscript.

3- In principle, a network could learn the stimulus/reward association by weakening connections to non-rewarded stimuli. Have the authors considered this possibility?

Good point, we thought about this possibility. When we increase the initial PC connectivity in our model (see Figure 2 below, Suppl. Fig. S7), the model can express a mixture between strengthening and weakening of the appropriate connections (panel a). However, this weakening of connectivity does not lead to an overall increase in selectivity for the vertical-rewarded- stimulus (see tuning curves, panel e). To see an effect from the weakening in the tuning, however, it would require stronger initial responses to non-preferred orientations and high baseline firing rate of the PCs. This would mean that initially pyramidal cells are broadly tuned and more strongly and promiscuously connected to each other. Learning the association then means to weaken connections from pyramidal cells tuned to non-rewarded orientations. However, this is not the case in our model as our pyramidal cells are sharply tuned and have low firing rates towards their non-preferred stimuli as observed in layer 2/3 (Niell and Stryker, Highly Selective Receptive Fields in Mouse Visual Cortex, Journal of Neuroscience 2008, Ko et al. "The emergence of functional microcircuits in visual cortex" Nature, 2013).

Figure 2 Spiking model with stronger initial recurrent connectivity. *a*: Evolution of excitatory connections (initially sampled from $\mathcal{N}(.01, .1)$). *a-e* as in the other figures. (Figure taken from Suppl. Fig. S7). The model displays a mixture of increase and decrease of synaptic strength (*a* – purple vs green), but does not show sharpening of the responses as firing rates to other stimuli are already low (vertical, *e*).

4- The connectivity between neuron groups in the model is all-to-all. As the authors' goal is to use experimental data to guide their theoretical framework, why wasn't the differential connectivity between neuron groups implemented in the model? Pfeffer et al, 2013, which the authors cite, provides information regarding connection probability within and between neuron groups. Connection probability between pyramidal neurons as well as between inhibitory neurons and pyramidal neurons is also widely reported. Differential connectivity between neuron groups could also expand the parameter space for the stimulus/ reward association.

That's an excellent suggestion. We therefore changed the connectivity in the model to the connectivity reported by experimental studies. We used the data from Pfeffer et al. 2013 for the connection probabilities and connections strengths between inhibitory neurons and from inhibitory neurons to excitatory neurons. Pfeffer et al. does not include data for connections from pyramidal cells to other cell types. For this, we used data from Jiang et al. 2016, Hofer et al. 2011, Pala and Petersen 2015, and Jouhanneau et al. 2018. Our model, now better constrained by experimental data, yields similar results (see Fig 2-5 of the main paper).

For connections between excitatory populations, we did not constrain those in the main paper. Neurons with dissimilar tuning may, however, not at all be connected to each other (Ko et al. "The emergence of functional microcircuits in visual cortex" Nature, 2013). We hence wanted to test how connectivity between PCs that is constrained to PCs with similar tuning affects the results (i.e. PCs with very dissimilar tunings have no synapses between them). We found that the extent of the increased representation to the rewarded stimulus is limited to connected PC populations (see Figure 3 below and Suppl. Fig. S6). Our model with differential PC connectivity hence yields a similar effect as reported by Goltstein et al. 2013. where only neurons with a preferred orientation similar to the rewarded one broadened their tuning curve towards that stimulus.

Figure 3 Model in which PCs are connected according to the cosine similarity of their preferred orientation, i.e. the connection probability of two cells is the cosine of the differences of their preferred angles. Panels as in the other figures. The horizontal population does not change its tuning as it is not connected to the vertical population (d,e). The angled populations are connected to the vertical population with a probability of 0.7. Thus, those populations can increase their tuning to the vertical stimulus (Figure taken from Suppl. Fig. S6).

Minor concerns:

Some of the terminology used in the manuscript is unclear.

1- In the introduction the authors state: "We hypothesized that the inhibitory circuit in L2/3 mediates the top down instructions to guide slow plastic changes..." What would the "slow plastic changes" be?

We were referring to changes in the excitatory connectivity. We referred to them as slow as they may require more time than available during the rewarded phase. We removed 'slow' from the sentence.

2- At the end of the introduction the authors state: "Our model offers testable predictions on the activity of different cell types during and after reward presentation". This is actually not what the model does. The model makes predictions about functional interactions between neuron groups, but does not provide any prediction in regard of what the actual activity of the different cell types.

We agree that the model does not make quantitative predictions for firing rates. The model, however, makes predictions about relative changes in firing rates for each of the cell types. We made this clearer by adding tables (see Tables S1 and S2) with the model assumptions, the experimental data constraining the model as well as the predictions from the model.

3- In the results: "Activity dependent plasticity then increases the connections between SST which are tuned to the rewarded stimulus and PV". What do the authors mean with "increases connections"? Would this be increase in number of connections, or increased functional connectivity, or increased synaptic strength?

Sorry about the confusion. We meant, the synaptic strength increased. We replaced 'connections' with 'connection strengths' to be more accurate.

Reviewer #3:

In this manuscript, Wilmes and Clopath propose a model of V1 microcircuitry that produces reward related sensory plasticity. The proposed model produces interesting dynamics during and after learning. It also takes into account the three main interneuron types observed in V1, which is commendable. However, I have major concerns with this manuscript. The most important one is that the interesting aspect of this model (i.e. two-stage learning) is highly counterintuitive and is contradicted by available experimental data (# 1 below). The remaining aspect of the model (SST to PV plasticity) is what speaks to the bulk of experimental data showing reward-related sensory plasticity. However, the model's explanation for this literature is just obtained by design due to some restrictive assumptions. On a related note, assumptions of the model are often not explicitly laid out. Indeed, some crucial assumptions are not experimentally supported, either due to lack of evidence or evidence to the contrary. Overall, I think the manuscript needs significant work to advance the field.

We now explicitly lay out the model's assumptions in a table (Table S1). We respond to the remaining points in more detail below.

1. A key assumption/prediction of the model is that stimulus selectivity in PCs is increased during the refinement phase in addition to the rewarded phase. In more behaviorist terms, even after extinction of a CS-US association, stimulus selectivity for a previously rewarded CS will increase. In addition to being a highly counterintuitive falsifiable prediction for primary sensory cortices, available data directly contradict this (Bieszczad and Weinberger, Extinction reveals that primary sensory cortex predicts reinforcement outcome, *Eur. J. Neurosci.*, 2012). A related falsifiable prediction that is not explicitly laid out is that recurrent connections between excitatory neurons do not strengthen during the rewarded phase (Fig 2f). This predicts that sensory plasticity in the presence of reward (i.e. almost all related experimental data) occurs without any changes in recurrent connections. This aspect has not been directly tested as far as I know, but seems quite counterintuitive.

*Extinction is a very interesting phenomenon. The cited study by Bieszczad and Weinberger shows that rats have a similar sized area representing a previously rewarded frequency after extinction (occurring after 3 days) compared to control. On the contrary, animals that learned an association between a frequency and a reward show an increased representation of that frequency. Hence, the change in representation seems to vanish with extinction. This study however does not reveal whether the original synaptic changes are being reversed. A study in the amygdala (Grewe et al. "Neural ensemble dynamics underlying a long-term associative memory", *Nature*, 2017) shows that the change in representation with extinction is not a reversal of the changes caused by association learning. Indeed, this study makes another interesting point: it shows that the changes in CS+ representation mostly happen in a consolidation phase a day after learning as opposed to during learning. Therefore, the presentation of the CS+ alone does not immediately lead to extinction, but on the contrary changes are still reinforced. Therefore, although counterintuitive, the experimental data seems to point to a similar direction as our model. While it is undoubtedly an interesting question of what the neural correlate of extinction is, it is beyond the focus of our study. Our model does not make predictions for what happens after extinction of a CS-US association. Extinction was not reported by Poort, Khan et al. ("Learning Enhances Sensory and Multiple Non-sensory Representations in Primary Visual Cortex", *Neuron*, 2015), when they alternated between visual and olfactory blocks. In the olfactory blocks, the visual stimuli were uninformative about reward. Hence, the vertical grating was not always paired with a reward. We now include a discussion of how our model relates to extinction in the discussion section 'Open questions'.*

We would like to clarify that the main role we propose for the inhibitory structure is that it allows for excitatory connections to continue changing after the reward. This does not exclude that changes in the excitatory structure already happen during the rewarded phase. Indeed, in the appendix, we showed that excitatory recurrent connections can change already in the presence of the reward. We now state that excitatory and inhibitory structures can co-develop more explicitly in the manuscript. We added a figure

to the appendix (Figure S3), showing which parameters regulate the amount of co-development. However, with the parameter set constrained by the experimental data (Figures 2-3), the co-development is small and mostly the excitatory structure develops after the rewarded phase. We think that this seemingly counterintuitive prediction challenges the traditional paradigm of excitatory connections as the source of all memory, and therefore fits well into current research that addresses excitatory versus inhibitory engrams (Mongillo et al. 2018, Barron et al. 2017). We also added a discussion section “Inhibitory engrams” to address this point.

Once this aspect is removed, all that the current model says is that SST to PV synaptic potentiation causes reward-related sensory plasticity. In my opinion, this result does not advance the field much as it is obtained by design in this model: the authors assume that only SST to PV synapse is plastic in addition to recurrent synapses. More importantly, this one-stage model is the model that addresses the bulk of experimental data as most experimental data only relate to the “rewarded phase” of this work. Overall, I find that for established experimental findings, the model presented here is trivial. And the interesting aspects of the model are either experimentally invalidated in case related data are available, or highly counterintuitive and unlikely to be verified.

2. It is assumed that only connections from SSTs to PVs and recurrent connections between PCs show synaptic plasticity. SST to PC or VIP to SST or PV to PC strengths are assumed to be fixed. What evidence is there to support these assumptions? Most importantly, what evidence is there to assume that SST to PV synapses show plasticity but not SST to PC synapses? My guess is that plasticity at SST to PC synapses will affect the refinement process mentioned here. Further, PV to PC synapses have been shown to undergo STDP (Vickers et al., Parvalbumin-Interneuron Output Synapses Show Spike-Timing-Dependent Plasticity that Contributes to Auditory Map Remodeling, Neuron, 2018). How does this affect the model, especially considering that PV cells may also exhibit stimulus selectivity?

We previously limited the plasticity to those connections to illustrate the key component underlying our modelling results. We agree that it is important to show that the plasticity of other connections in the network do not compromise the mechanism. Therefore, we added plasticity to all the remaining connections. For example, the PV to PC plasticity was based on the suggested experimental data of Vickers et al. (see supplementary Figure S4 and Figure 4 below). All our results hold in these new simulations (see Figure 5 below for simulations with plasticity at all connections and sharply tuned PVs to address the following point (3) of the reviewer).

Simulating our model with plasticity at all synapses, we observe that plasticity of PV-to-PC synapses (as reported by Vickers et al.) strengthens connections from PVs to PCs (see Figure 4 below, panel g), because PVs are driven by PCs and hence follow the PCs in firing. This strengthens bidirectional PC-PV interactions, which are in line with recent experimental data (Znamenskiy et al. “Functional selectivity and specific connectivity of inhibitory neurons in primary visual cortex”, bioRxiv 2018). The stronger the PV-to-PC inhibition, the stronger the disinhibitory effect of the SST-to-PV structure, and hence the stronger the

development of the excitatory structure during the refinement phase (also see Figure 7b further below). Plasticity of SST-to-PC connections does not affect the refinement process.

Figure 3 Model with all connections plastic yields similar results. A-f as in the other figures. G shows the evolution of synaptic weights over time from SST-to-PC (left) and PV-to-PC (right) (Figure taken from Suppl. Fig. S4).

The mechanism still works if PVs are stimulus-selective (see Figure 5 below).

Figure 5 Model with tuned PVs and plasticity on all connections. a-e as in the other figures. f: Rasterplot from the beginning of the simulation illustrating that PVs are sharply tuned to orientation in this model.

3. The authors assume that PV neurons show broad stimulus-untuned responses. While some literature does exist supporting this assumption (as the authors cite), a lot of papers show spatial tuning for PV neurons. For example, Cardin et al. show that fast-spiking interneurons exhibit clear stimulus-selectivity (Cardin et al., Stimulus Feature Selectivity in Excitatory and Inhibitory Neurons in Primary Visual Cortex, J. Neurosci. 2007); numerous papers show that most parvalbumin neurons are fast-spiking (e.g. Hu et al. Interneurons. Fast-spiking, parvalbumin⁺ GABAergic interneurons: from cellular design to microcircuit function, Science, 2014). Additional papers also support spatial tuning in genetically identified PV neurons (Runyan et al., Response features of parvalbumin-expressing interneurons suggest precise roles for subtypes of inhibition in visual cortex, Neuron 2010; Wilson et al., Division and subtraction by distinct cortical inhibitory networks in vivo, Nature 2012).

Thus, this fundamental aspect of their model seems to be at odds with considerable experimental data.

We do not make assumptions about the spatial tuning of PVs, as this does not influence the mechanism that we propose. We assumed that PVs in layer 2/3 are broadly tuned to orientation, not space, based on several studies (Hofer et al. 2011, Atallah et al. 2012; GAD65/67: Liu et al. 2009, Kerlin et al 2010, Niell and Stryker 2008, Sohya et al. 2007). We now added a more in-depth review of the existing literature in the discussion section of the manuscript ('Orientation tuning of interneurons'). While there is one study showing sharply tuned PVs in layer 2/3 (Runyan et al. 2010), the majority of studies report that interneurons are less sharply tuned to orientation than excitatory cells (Hofer et al. 2011, Atallah et al. 2012, Liu et al. 2009, Kerlin et al 2010, Niell and Stryker 2008, Sohya et al. 2007).

Additionally, we now show that our results do not depend on the tuning of PVs by simulating a network with tuned PVs (see Figure 6 below and Suppl. Mat. Fig. S5).

The study by Cardin et al., which was mentioned by the reviewer, shows that FS cells are only a bit less tuned than regular spiking (RS) cells in layer 4, but these are cells in layer 4 and not in layer 2/3. This study also shows that RS cells in layer 4 are more sharply tuned than in layer 2/3, which are in turn more sharply tuned than fast spiking (FS) cells in layer 2/3.

Figure 6 Model with sharply tuned PVs. a-e as in the other figures. f: The rasterplot from the beginning of the simulation demonstrates that PVs are now sharply orientation-tuned. (Figure taken from Suppl. Figure S5.)

4. SSTs inhibit VIPs equally strongly as they inhibit PCs (Pfeffer et al. 2013). This connection is not included in the model. How do results depend on this connection?

This is a very good point. In order to test that, we constrained our connectivity according to experimental data. To this end, we used the data from Pfeffer et al. 2013 for the connection probabilities and connections strengths between inhibitory neurons and from inhibitory neurons to excitatory neurons. As Pfeffer et al. does not include data for connections from pyramidal cells to other cell types, we used data from Jiang et al. 2016, Hofer et al. 2011, Pala and Petersen 2015, and Jouhanneau et al. 2018 for these connections. We can, therefore, say that the results do not depend on the SST-to-VIP connection (see Figures 2-5 of the manuscript). After including the SST-to-VIP connection into the model, and VIPs still suppress SSTs. This is in line with the experimental observation that VIPs potently inhibit SSTs in several cortical areas (e.g. Williams et al. "Higher-Order Thalamocortical Inputs Gate Synaptic Long-Term Potentiation via Disinhibition", Neuron 2019).

5. PV to PC connection strength is approximately twice that of SST to PC strength (Pfeffer et al. 2013). In this model, the strengths are assumed to be 0.25 and 0.3, respectively, which is at odds with experimental data. My guess is that increasing the PV to PC strength to reflect experimental data will reduce the excitatory structure index during refinement.

This is again a good point. As we mentioned below, we now constrained the connectivity in our model by the experimental data (Pfeffer et al. 2013). The PV-to-PC connection strength is now .55nS (SST-to-PC strength is still .3nS).

Effect of increasing PV to PC strength: To the contrary, increasing the PV to PC strength increases the excitatory structure index during refinement. An increased PV-to-PC connectivity increases the strength of the SST-PV-PC pathway and hence the disinhibition of PCs during the rewarded stimulus in the refinement phase. To directly show the effect of the PV-to-PC connection, we simulated the model with different connection strengths and show that the stronger the PV-to-PC connection, the more the excitatory structure develops later during the refinement phase (See Figure 7 below). The PV-to-PC connection strength is now .55nS (SST-to-PC strength is still 0.3nS).

Figure 7 The excitatory structure develops to lesser extent during the rewarded phase with stronger PV-PC connections (a). It develops more during the refinement phase with stronger PV-PC connections (b). a-b: Excitatory structure index (see methods) as a function of PV-to-PC strength at the end of the rewarded phase (a) and at the end of the refinement phase (b) c: Example simulation with PV-to-PC = 0.65nS, panels as in the other figures.

6. The interesting aspects of the current model primarily work via disinhibition of PCs by SSTs. However, data show that the primary effect of SSTs on PCs is inhibition, not disinhibition (e.g. Lee et al. Activation of specific interneurons improves V1 feature selectivity and visual perception, Nature 2012). Is the net effect of SSTs on PCs inhibition or disinhibition in the model, especially after #5 above is addressed?

We apologise for the lack of clarity. The net effect of SSTs on PCs is inhibitory (see Figure 8 below), consistent with experimental data of e.g. Lee et al. 2012 (see Figure below). The inhibitory structure between SSTs and PVs yields that PCs are less inhibited (disinhibited) during the rewarded stimulus than during other stimuli.

Figure 8 The net effect of SSTs on PCs is inhibitory. We took the network connectivity at the end of the refinement phase (during which PCs are selectively disinhibited), we then drove the PCs with layer 4 input with a connection strength of 0.4nS. We then simulated the network for 1.4s (control) with all other settings as before, and for 1.4s with completely suppressed SSTs. We compared the PC activity in a control case with that during complete suppression of SSTs. Suppressing SSTs leads to a net increase in PC activity, showing that the net effect of SSTs is inhibitory. Left: Average activity of all populations to all stimuli in the control case and with all SSTs suppressed. Right: Tuning curves for the four PC populations in the control case (black) and with SSTs suppressed (orange).

7. How do the authors account for a reduction in stimulus selectivity for unrewarded stimuli as was observed by Poort et al. 2015?

In Poort et al. 2015, the selectivity to both stimuli increased and the sensory representation of the rewarded stimulus was enhanced as in our model. See Poort et al. Suppl. Fig. S3 C for percentages of cells increasing and decreasing their responses. 35% increase their response to both stimuli, 32% decrease their

response to both, and 8% both increase their response to the preferred and decrease their response to the non-preferred stimulus.

8. The authors may want to cite Kilgard and Merzenich, Cortical map reorganization enabled by nucleus basalis activity, Science, 1998.

Good idea! We cited in the discussion section on 'Top-down signals'.

9. Reward is not a behaviorally relevant "context". It is a stimulus.

We agree that 'context' is an ill-defined term. We treat reward as a stimulus in our model and therefore refer to it as a stimulus now.

10. "We wanted to test whether interneurons can learn from a top-down signal." This statement implies that an experimental test is conducted in the present study and is thus misleading. It should instead say something like, "Based on our current experimental understanding, we wanted to test whether interneurons can learn from a top-down signal in a model of cortical circuitry."

We thank the reviewer for the suggestion. We changed the sentence to make sure the reader understands that we conduct computational and not experimental work.

Reviewers' Comments:

Reviewer #1:

Remarks to the Author:

The authors have made changes to the manuscript that have addressed my major concerns. While there are still features of the model that are speculative, the model generates specific predictions that are of relevance for the neuroscience community, so overall I am supportive of publication of this article.

Reviewer #2:

Remarks to the Author:

The revised manuscript has addressed all of my concerns. The additional data and text edits have strengthened the support for the authors conclusions. I do not have additional comments.

Reviewer #3:

Remarks to the Author:

In general, the authors have been very responsive to my comments and have done an excellent job at revising the manuscript. So I am in favor of publication. However, I still have one concern that would be important to clarify, though I leave this to the authors' discretion.

Regarding extinction, my main point was that extinction might indeed be an "unlearning" for the *primary sensory cortices*. There are plenty of data showing that behaviorally and electrophysiologically in higher order cortical areas or subcortical structures, extinction is not unlearning, but is new learning. The Grewe et al. paper is one among many such papers. However, I do not really find it appropriate to extrapolate from these data to the primary visual cortex. Even though the authors show that there is some change in excitatory structure during the rewarded phase, the refinement phase (this is behavioral extinction, whether or not neurally it acts as extinction) produces 1-2 order(s) of magnitude change in the excitatory structure (Fig S3). So the authors aren't simply saying that changes during the reward phase get maintained once the reward is removed, but that they change by 1-2 orders of magnitude. I still maintain that this is highly counter-intuitive for primary sensory cortices given existing data. Thus, I think this should be highlighted as a major falsifiable prediction of the model. In general, I also think there should be a list of falsifiable predictions from the model that can be tested experimentally. Digging through the paper for this list would not be necessarily easy for experimentalists.

Also as a side-note, the experiment by Poort, Khan et al. is not behavioral extinction. Behavioral extinction typically requires no reward to be delivered, as unpredictable rewards often act to induce spontaneous recovery of reward seeking behaviors. So this is not an appropriate test of the model.